# Distinct responses to rare codons in select *Drosophila* tissues

**Scott R Allen[1], Rebeccah K Stewart[2], Michael Rogers[2], Ivan Jimenez Ruiz[3], Erez Cohen[1], Alain Laederach[3], Christopher M Counter[2], Jessica K Sawyer[2], Donald T Fox[1,2]\***

[1]Department of Cell Biology, Duke University, Durham, United States; [2]Department of Pharmacology and Cancer Biology, Duke University, Durham, United States; [3]Department of Biology, University of North Carolina at Chapel Hill, Chapel Hill, United States

**Abstract** Codon usage bias has long been appreciated to influence protein production. Yet, relatively few studies have analyzed the impacts of codon usage on tissue-specific mRNA and protein expression. Here, we use codon-modified reporters to perform an organism-wide screen in *Drosophila melanogaster* for distinct tissue responses to codon usage bias. These reporters reveal a cliff-like decline of protein expression near the limit of rare codon usage in endogenously expressed *Drosophila* genes. Near the edge of this limit, however, we find the testis and brain are uniquely capable of expressing rare codon-enriched reporters. We define a new metric of tissue-specific codon usage, the tissue-apparent Codon Adaptation Index (taCAI), to reveal a conserved enrichment for rare codon usage in the endogenously expressed genes of both *Drosophila* and human testis. We further demonstrate a role for rare codons in an evolutionarily young testis-specific gene, *RpL10Aa*. Optimizing *RpL10Aa* codons disrupts female fertility. Our work highlights distinct responses to rarely used codons in select tissues, revealing a critical role for codon bias in tissue biology.

**\*For correspondence:**
don.fox@duke.edu

**Competing interest:** The authors declare that no competing interests exist.

## Editor's evaluation

This report is significant in providing strong evidence that differences in codon optimality in mRNAs can underlie tissue-specific differences in expression in fruit flies and that this mechanism is at play in restricting expression of an evolutionarily young ribosomal protein gene with higher than average rare codon usage to the testis versus ovaries in a manner critical for female fertility. The work breaks new ground in identifying codon usage as a basis for tissue-specific gene expression in animals.

## Introduction

The genetic code is redundant, with 61 codons encoding only 20 amino acids (*Crick, 1968*; *Zuckerkandl and Pauling, 1965*). It was initially thought that synonymous substitutions, those leading to changes in nucleotide sequence but resulting in an identical protein sequence, were functionally 'silent.' However, it is now clear that this is not the case. Synonymous codons are used at varying frequencies throughout a given genome (*Grantham et al., 1980*; *Ikemura, 1985*; *Sharp and Li, 1986*). This disproportionate usage frequency among synonymous codons is termed codon usage bias (hereafter: codon bias).

Codon bias is governed by several biochemical mechanisms and has diverse biological consequences. In general, mRNAs enriched in codons commonly used in a given species are more stable and are more robustly translated (*Presnyak et al., 2015*; *Sorensen and Pedersen, 1991*; *Yan et al.,*

*2016*; *Yu et al., 2015*; *Zhao et al., 2017*). Conversely, a high frequency of rare codons in an mRNA can cause ribosomal stalling and trigger RNA degradation or premature translation termination (*Buschauer et al., 2020*; *Radhakrishnan et al., 2016*; *Yang et al., 2019*). Codon bias impacts expression and structure of clock proteins that underlie the in vivo circadian clock function in *Neurospora*, cyanobacteria, and *Drosophila* (*Fu et al., 2016*; *Xu et al., 2013*; *Zhou et al., 2013*), protein secretion in yeast (*Pechmann et al., 2014*), and virus/host interactions (including in COVID-19, *Alonso and Diambra, 2020*; *Shin et al., 2015*).

Emerging studies suggest codon bias plays an important role in fundamental tissue-level processes. For example, codon bias underlies differences between maternal and zygotic mRNAs in developing zebrafish, *Xenopus*, mouse, and *Drosophila* (*Bazzini et al., 2016*). Further, while rare codons generally destabilize mRNAs in *Drosophila* whole embryos, this effect is attenuated within the embryonic central nervous system, where codon bias has little impact on mRNA stability (*Burow et al., 2018*). In humans, tRNA levels have been shown to differ between tissues and cell types, impacting the translation efficiency of rare codon-enriched transcripts (*Dittmar et al., 2006*; *Gingold et al., 2014*), and RAS isoforms that differ in codon usage have different transcription/translation kinetics based on cellular context (*Fu et al., 2018*). Studies analyzing gene expression datasets cross-referenced with codon usage also suggest that the impact of codon bias on protein expression may differ between tissues. For example, codon usage frequencies differ between tissue-specific gene sets in numerous plant species (*Camiolo et al., 2012*; *Liu, 2012*), *Drosophila* (*Payne et al., 2019*), and potentially humans (*Plotkin et al., 2004*; *Sémon et al., 2006*), hinting that codon usage could play a fundamental role in tissue and cellular identity.

Here, we leverage the genetic and cell biological strengths of *Drosophila melanogaster* to reveal tissue-specific impacts of codon bias. Using a library of codon-altered reporters, we conduct an organism-wide screen during *Drosophila* development. We find that reporter protein expression declines drastically over a narrow range of rare codon usage in a gene. We further show that specific tissues, namely the testis and brain, are distinct in the ability to robustly express proteins encoded by rare codons. Focusing further on the testis, the tissue with the strongest protein expression from rare codon-enriched genes, we find the male germ cells and somatic hub cells are capable of robust rare codon-derived protein production, while somatic cyst cells are not. By developing a new metric to examine tissue-specific codon usage, tissue-apparent CAI (taCAI), we find that both *Drosophila* and human testes express genes that are enriched in rare codons relative to other tissues. Examining the physiological significance of this conserved enrichment, we highlight a role for abundant rare codons in *RpL10Aa*, an evolutionarily young gene encoding a testis-specific ribosomal subunit. We present evidence that rare codons in this gene are critical for fertility. Just as chromatin states regulate tissue-specific gene expression at the level of transcription, here we find a biologically significant role for codon bias, an important determinant of mRNA translation, in regulating tissue-specific protein production.

## Results

### A cliff-like protein expression threshold determined by rare codons in *Drosophila*

We previously showed that severely altering codon content impacts protein expression from *RAS* family genes or from a *GFP* reporter in *Drosophila melanogaster* and humans (*Ali et al., 2017*; *Lampson et al., 2013*; *Peterson et al., 2020*; *Sawyer et al., 2020*). This prompted us to systematically determine the impact of codon usage on protein expression. To do so, we again used phiC31 integrase for stable site-specific insertion of a single copy of transgenic codon-modified *GFP* reporters (*Supplementary file 1*) into the *attP40* locus on chromosome 2. This locus is well-established to express transgenes at robust and reproducible levels (*Markstein et al., 2008*). As we have done previously (*Sawyer et al., 2020*), we placed all of our transgenic reporters in the same vector backbone, which uses a *ubiquitin* (*ubi-*) promoter sequence for robust gene expression (Methods). In this way, we control for the effects of chromatin environment on gene expression and ensure that any differences in protein production are due to differences in coding sequence.

We took two approaches to introduce rare codons into our *GFP* reporters. In the first approach, we used a random number generator to select positions in the *GFP* coding sequence to engineer rare

codons. This approach keeps the majority of rare codons dispersed throughout the *GFP* sequence (*Figure 1A*). We generated 10 reporters using this dispersed codon strategy, and such reporters are designated with a 'D.' Given the potential importance of contiguous stretches of rare codons (*Chu et al., 2014*; *Hayes et al., 2002*; *Kramer and Farabaugh, 2007*; *Spanjaard and van Duin, 1988*), we also employed a second approach in which all rare codons were clustered at either the 5' or 3' end of the coding sequence. We generated six reporters using this clustered codon strategy, and all reporters generated this way are designated 'C5' or 'C3.' In defining rare codons, we referred to the Kazusa codon database (*Nakamura et al., 2000*). We identified the most used synonymous codon in the *Drosophila melanogaster* genome as the 'common' codon, the least used codon as the 'rare' codon, and any additional synonymous codons between the rare and common codon as 'middle' codons. We designed reporters to contain a specific percentage of rare codons, leaving the remaining portion of the coding sequence split between common and middle codons (*Figure 1A*). These reporters were all designed using an identical eGFP amino acid sequence to rule out any effects of amino acid bias on protein production levels (*Weber et al., 2020*). All reporters are designated with the fluorophore (e.g. GFP) followed by the percentage of rare codons, followed by a 'D' or 'C' designation (e.g. *GFP50D*, *GFP70C3'*).

As a convenient method to screen how codon substitutions in GFP impact protein expression across the entire animal, we imaged wandering third instar larvae (WL3) for each transgenic reporter on a fluorescent dissection microscope (**Methods**). These animals contain fully formed organ systems and are translucent to facilitate imaging. We used the *GFP0D* reporter with no rare codons as a baseline for comparison. Upon examining animals for each reporter, we noticed an apparent 'all or none' fluorescence pattern (*Figure 1B–Q*). To quantitate our observations seen by fluorescent microscope, we measured protein production by western blot (*Figure 1R*, *Figure 1—figure supplement 1*). For reporters with dispersed rare codons, 2/2 reporters with up to 50% rare codons display ubiquitous fluorescence throughout the animal, similar to *GFP0D*. These two reporters (*GFP30D* and *GFP50D*) produce animal-wide GFP protein at levels greater than or equal to 50% of GFP0D (*Figure 1B–D and R*, *Figure 1—figure supplement 1A*). In contrast, 7/7 reporters with greater than 50% dispersed rare codons display no visible fluorescence in the animal and produce protein at a level less than 0.3% of GFP0D (*Figure 1B, E–K and R*, *Figure 1—figure supplement 1B,C*). Similarly, 6/6 reporters with 50% or more clustered rare codons display no fluorescence (*Figure 1L–Q*). Two of these six clustered codon reporters, *GFP50C5'* and *GFP60C3'*, produce barely detectable protein levels by western blot, between 7–9% that of GFP0D, whereas the other four reporters yield less protein than 1% of GFP0D (*Figure 1R*, *Figure 1—figure supplement 1C,D*). To assess whether mRNA abundance correlates with protein abundance for each reporter, we performed quantitative RT-PCR in each transgenic GFP line and normalized the mRNA abundance to that of GFP0D (*Figure 1—figure supplement 1S*). Overall, these values show a consistent trend between mRNA and protein abundance for each reporter (*Figure 1R vs. S*), suggesting that inclusion of rare codons impacts both protein and RNA abundance for these reporters.

While it is not surprising that there would be a rare codon threshold beyond which proteins are no longer produced, what is surprising is the magnitude of the differences in protein production (from robust to barely/not detectable) over a narrow range of rare codon usage. A 50% increase in rare codon usage between *GFP0D* and *GFP50D* causes a 50% reduction in protein levels. Yet an additional 10% increase in rare codon usage between *GFP50D* and *GFP60Dv1* decreases detectable protein by 99.5% (*Figure 1R*, *Figure 1—figure supplement 1A*).

To determine if the steep decline in protein levels between 50 and 60% dispersed rare codons is robust and reproducible, we independently designed two more 60% dispersed rare codon reporters. These reporters, *GFP60Dv2* and *GFP60Dv3*, contain rare codons that differ in sequence/position from *GFP60Dv1*. All three GFP60D reporters show the same fluorescence pattern (no detectable GFP), and when protein levels are compared to GFP50D by western blot we again observe a decrease in protein levels of at least 99.5%, with a 99.85 decrease on average across all three GFP60D reporters (*Figure 1E–G and R*, *Figure 1—figure supplement 1B*). Overall, our results indicate there is a narrowly defined window of rare codon content in which protein production drops from robust to barely detectable/none.

We next put our observed cliff-like decline in protein levels from reporters in context with endogenous *Drosophila melanogaster* gene sequences. To look at the codon usage of endogenous genes, we

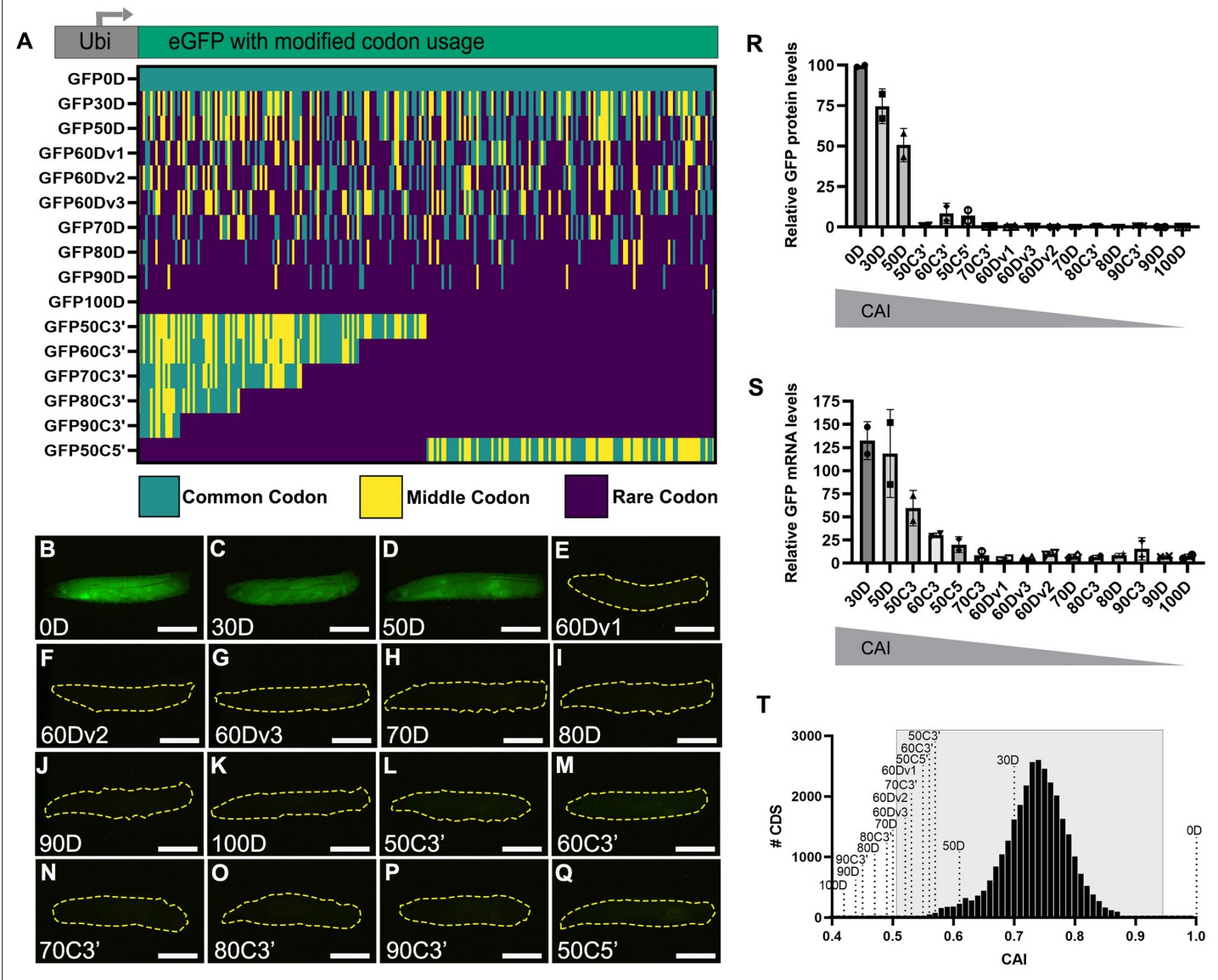

**Figure 1.** Strictly defined limits on rare codon usage regulate gene expression. (**A**) Top- schematic of each indicated reporter. Bottom- heat map indicating codon usage across each reporter coding sequence. Color key indicates codon usage along the length of the coding sequence. (**B–Q**) Representative fluorescent images taken under identical camera exposure settings of live male wandering third instar larvae (WL3) containing stable genomic insertions of the indicated *GFP* reporter. Scalebars are 1 mm. Yellow dashed outlines highlight the larvae in images where no GFP signal was apparent. (**R**) Western blot protein quantifications of indicated reporters normalized to total protein stain then plotted as percentages relative to GFP0D for whole third instar larvae (two replicates, N=5 each, plotting individual data points and mean ± SEM). Reporters listed in descending order by Codon Adaptation Index (CAI). See *Figure 1—figure supplement 1* for blot images. (**S**) mRNA abundances of the indicated reporters normalized as percentages relative to GFP0D for whole third instar larvae (two replicates, N=5 each, plotting individual data points and mean ± SEM). (**T**) Histogram of CAI values for each transcript in the *Drosophila melanogaster* genome. Reporter CAI values are indicated with dotted lines. Full range of endogenous genes highlighted in light gray box.

The online version of this article includes the following source data and figure supplement(s) for figure 1:

**Figure supplement 1.** Western blots of whole wandering third instar larvae (WL3).

**Figure supplement 1—source data 1.** Raw data file and associated PDF figure of uncropped blot presented in *Figure 1—figure supplement 1A,B* and quantified in *Figure 1R*.

**Figure supplement 1—source data 2.** Raw data file and associated PDF figure of uncropped blot presented in *Figure 1—figure supplement 1A,B* and quantified in *Figure 1R*.

**Figure supplement 1—source data 3.** Raw data file and associated PDF figure of uncropped blot presented in *Figure 1—figure supplement 1C* and quantified in *Figure 1R*.

*Figure 1 continued on next page*

*Figure 1 continued*

**Figure supplement 1—source data 4.** Raw data file and associated PDF figure of uncropped blot presented in *Figure 1—figure supplement 1C* and quantified in *Figure 1R*.

**Figure supplement 1—source data 5.** Raw data file and associated PDF figure of uncropped blot presented in *Figure 1—figure supplement 1D* and quantified in *Figure 1R*.

**Figure supplement 1—source data 6.** Raw data file and associated PDF figure of uncropped blot presented in *Figure 1—figure supplement 1D* and quantified in *Figure 1R*.

**Figure supplement 2.** Codon Adaptation Index (CAI) is not correlated with gene length.

turned to the well-established Codon Adaptation Index (CAI) as a metric for measuring codon usage optimality (*Sharp and Li, 1986*). CAI calculates the degree of optimal codon usage in a transcript based on the usage frequency of that codon in a reference set of genes. Higher CAI scores indicate more common codon usage, where a score of 1 indicates a transcript only uses the most common codons for each amino acid (*Nakamura et al., 2000*). We plotted the CAI values of each individual transcript in the *Drosophila melanogaster* genome (*Figure 1T*, Methods). We similarly computed the CAI values for each of our codon-modified *GFP* reporters (*Figure 1T*).

Comparing CAI values between reporters and endogenous genes reveals a striking cliff-like protein expression limit imparted by rare codons. This limit is reflected in the all-or-none expression pattern of our reporters. The lowest CAI of any reporter (*GFP50D*, *Figure 1R*, *Figure 1—figure supplement 1*) with detectable fluorescence resides near the limit of CAI for endogenously expressed genes (only above the CAI of 2% of all expressed genes- *Figure 1T*, **see left boundary of gray rectangle**). Below the CAI value of *GFP50D*, none (0/13) of our reporters are expressed at the level of detectable fluorescence, and only two of these reporters produce protein detectable by western blot at low levels (*Figure 1R*, *Figure 1—figure supplement 1*). We note that we find no correlation between mRNA or protein length and CAI in *Drosophila* (*Figure 1—figure supplement 2*), in contrast to a previous study comparing frequency of optimal codons to protein length (*Duret and Mouchiroud, 1999*). Our findings suggest that for endogenous genes in *Drosophila*, a limit on robust rare codon-derived protein expression is in the range of 50–60% rare codon usage or between CAI scores of 0.61 and 0.57. This limit is experimentally validated by the failure of 13/13 transgenic reporters below this range to produce appreciable levels of protein (*Figure 1R*, *Figure 1—figure supplement 1*), and is reflected by the rarity of endogenous genes at or below this range (*Figure 1T*).

## The testis and brain robustly express rare codon-enriched reporters

While overall we observe a cliff-like threshold in reporter GFP levels (*Figure 1R*, *Figure 1—figure supplement 1*), none of the clustered rare codon reporters reside within our determined CAI-dependent limit to be detected by fluorescence. Given this, we generated another clustered codon reporter (*GFP54C3′*). In designing *GFP54C3′*, we clustered all of the rare codons in the 3′ end of the coding sequence and used only the most common or most rare codons for each amino acid. Compared to *GFP50D*, this design increased rare codon content while maintaining a higher overall CAI score (0.64, *Figure 2A*). *GFP54C3′* falls near the limit of CAI for endogenously expressed *Drosophila* genes, ranking higher than just 5% of endogenous transcripts. Unlike the all-or-none expression pattern of the reporters shown in *Figure 1*, GFP54C3′ shows no signal in most of the animal but has a robust GFP fluorescence in two distinct areas of the larva (*Figure 2C*). This suggests that distinct tissues respond differently to rare codons.

Given the unique tissue-specific pattern with this reporter, we generated additional reporters to further understand the sequence parameters driving this expression. To assess whether the tissue-specific expression of GFP54C3′ is due to an unknown sequence motif, we generated an entirely different reporter with similar sequence parameters. To do so, we generated a hybrid *mCherry/GFP* reporter. Both *mCherry* and *GFP* are similar in size (708 and 717 nucleotides, respectively). Since the rare codons in *GFP54C3′* are clustered towards the 3′ end, we designed *mGFP100Dv1*, where we redesigned *GFP100D* by creating a fusion protein linked to a 5′ *mCherry* sequence (Methods). This *mCherry* is naturally enriched in common codons, and the fusion to *GFP100D* creates a reporter with a CAI similar to *GFP54C3′* (*Figure 2B*). mGFP100Dv1 rescues the lack of expression seen with GFP100D in a tissue-specific manner (*Figure 2D* vs. *Figure 1K*). Using further altered versions of this

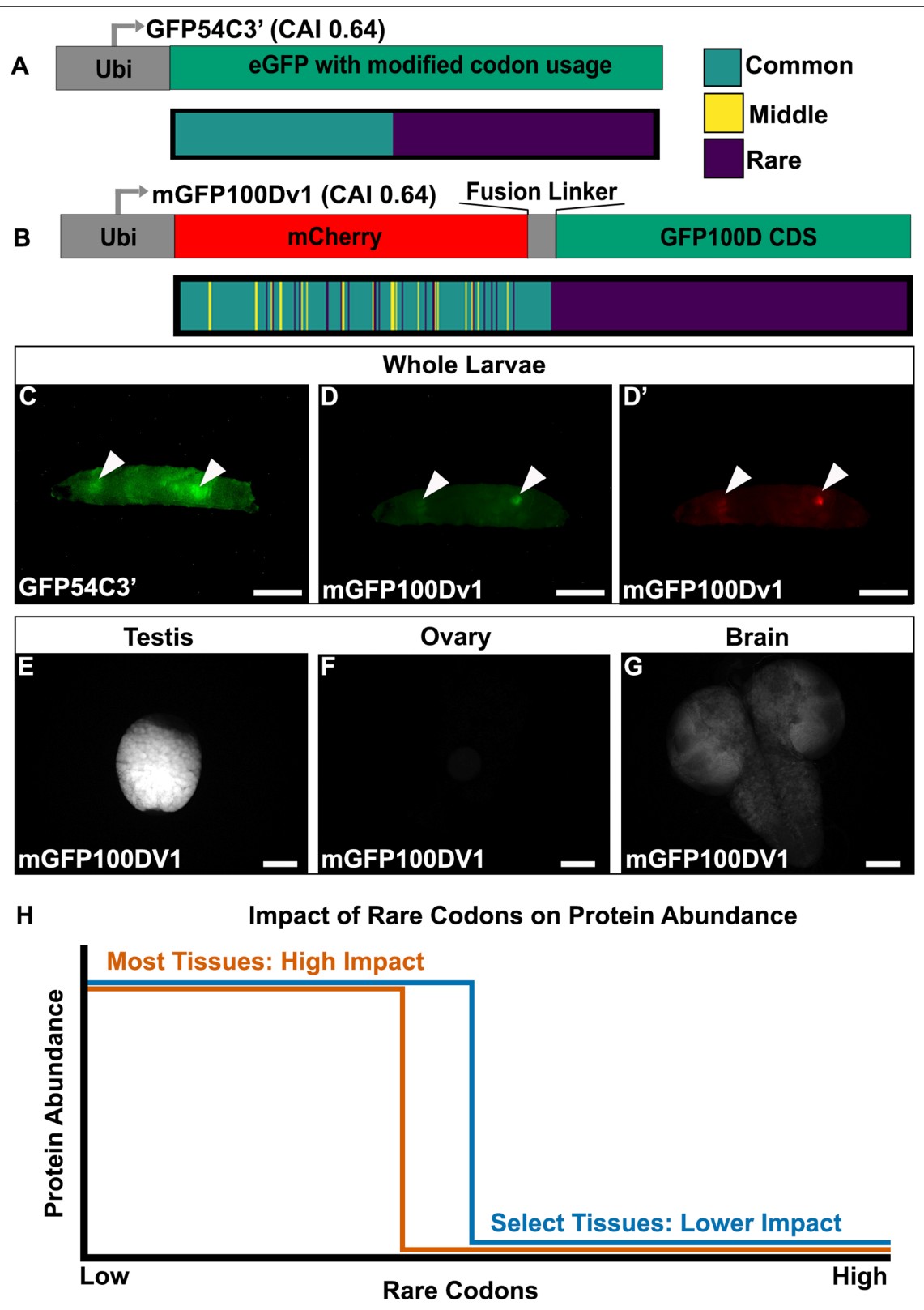

**Figure 2.** Tissues exhibit distinct responses to rare codons. (**A**) Top- reporter design and Bottom- codon usage (see key) along the CDS of *GFP54C3'*. CAI for *GFP54C3'* indicated in parentheses. (**B**) Top- reporter design and Bottom- codon usage along the CDS of *mGFP100Dv1* (see key in panel A). CAI for *mGFP100Dv1* indicated in parentheses. (**C**) Representative fluorescent images of live male wandering third instar larvae (WL3) with stable genomic insertion of *GFP54C3'*. (**D**) Representative fluorescent image of live male WL3 larva with stable genomic insertion of *mGFP100Dv1* taken at GFP

*Figure 2 continued on next page*

*Figure 2 continued*

excitation wavelength. (**D′**) Representative fluorescent image of live male WL3 larva with stable genomic insertion of *mGFP100Dv1* taken at mCherry excitation wavelength. Scalebars for (**C–D**)′ are 1 mm. (**E–G**) Representative fluorescent images of dissected *mGFP100Dv1* larval tissues. Images are taken under identical conditions with fluorescence intensity normalized to testis. Scalebars for (**E–G**) are 100 µm. (**H**) Conceptual depiction of tissue-specific differences in rare codon tolerance, as revealed by our reporters. As rare codons increase (moving to the right on the x-axis), protein abundance (y-axis) undergoes a cliff-like decline. However, the point at which select tissues hit the edge of the cliff is distinct.

The online version of this article includes the following figure supplement(s) for figure 2:

**Figure supplement 1.** Tissue-specific rescue of GFP100D is not dependent on mCherry or linker sequence.

hybrid mCherry/GFP reporter (*Figure 2—figure supplement 1A*, Methods), we ruled out that the tissue-specific rescue of expression is due to an unknown sequence motif in the mCherry or fusion linker (*Figure 2—figure supplement 1B–G*). We also examined the sequences of all reporters for co-occurring codon pairs known to impact translation fidelity, such as AGG and CGA (*Letzring et al., 2010*; *Spanjaard and van Duin, 1988*), and find no predictive pattern in their occurrence between our tissue-specific and non-tissue-specific reporters (*Supplementary file 2*).

Having established that tissue specificity can be achieved using multiple different coding sequences, we next determined which tissues express rare codon-enriched GFP54C3′ and mGFP100Dv1. In larvae, we dissected out the two bright tissues from mGFP100DV1. In doing so, we noticed that male larvae always have two bright tissues, whereas female larvae always have one. Our dissections revealed this difference to be because mGFP100DV1 expresses brightly in the larval testis (*Figure 2E*) but is not detectable in larval ovaries (*Figure 2F*). The other mGFP100DV1 bright tissue, in common to both female and male mGFP100DV1 larvae, is the brain (*Figure 2G*). Together, our results reveal that, within a narrow range of 50–60% rare codons, a select few tissues are capable of robustly producing protein from rare codon-enriched transcripts (*Figure 2H*).

We next turned to adult animals, to assess if the distinct testis and brain expression carries forward to this stage. We used a combination of fluorescence microscopy of isolated tissues (*Figure 3A–L′*) and quantitative western blotting (*Figure 3M*, Methods). This analysis again revealed that the testis and brain express mGFP100Dv1 with the testis having the strongest relative protein expression (*Figure 3E, F and M*). Further, we find that other tissues, such as the ovary and male accessory glands, do not express this reporter at a detectable level (*Figure 3G, H and M*). Conversely, common codon-enriched GFP0D is readily detectible, albeit at different cell type-specific levels in all adult tissues examined (*Figure 3A–D and M*, *Figure 3—figure supplement 1*). This suggests that lack of mGFP100Dv1 expression is not due to strong tissue-level differences in promoter strength between tissues.

To quantify the impact of rare codons on protein production in each tissue, we compared protein levels of mGFP100Dv1 to GFP0D from three replicate western blots (*Figure 3M*, *Figure 3—figure supplement 1*). There is no statistically significant difference in protein levels between GFP0D and mGFP100Dv1 for both testis and brain in adult males, consistent with these tissues having a distinctly higher tolerance for rare codons (*Figure 3M*). In contrast, both the ovary and accessory gland produce mGFP100Dv1 protein at levels less than or equal to 2% that of GFP0D protein (*Figure 3M*), consistent with a lower rare codon tolerance in these (and most other) *Drosophila* tissues (*Figure 2H*). While we do not observe differences in the ability of the brain to express rare codon-enriched reporters between sexes (*Figure 3—figure supplement 2*), the testis but not the ovary robustly expresses protein derived from rare codon-enriched reporters (*Figure 3M*). One possible explanation for the tissue-specific differences in protein expression could be differences in the strength of the *ubiquitin* promoter in each tissue. However, our fluorescence-based approach suggested this is not the case (*Figure 3A–D,*). We further assessed this by examining the levels of GFP0D (all common codons) by western blot in each tissue. Interestingly, the ovary expresses GFP0D at similar levels to the testis, yet rare codon-enriched mGFP100Dv1 is expressed 100-fold higher in the testis than in the ovary (*Figure 3—figure supplement 1B*). This confirms that promoter strength is not a confounding variable. A second confounding variable could be chromatin context. To assess if testis and brain specificity of an mGFP100D reporter still occurs when integrated at a different locus, we integrated a reporter with similar codon content (mGFP100Dv8) into the *attP2* locus on chromosome 3, and again observed the same testis and brain specificity of this rare codon-enriched reporter (*Figure 3—figure supplement*

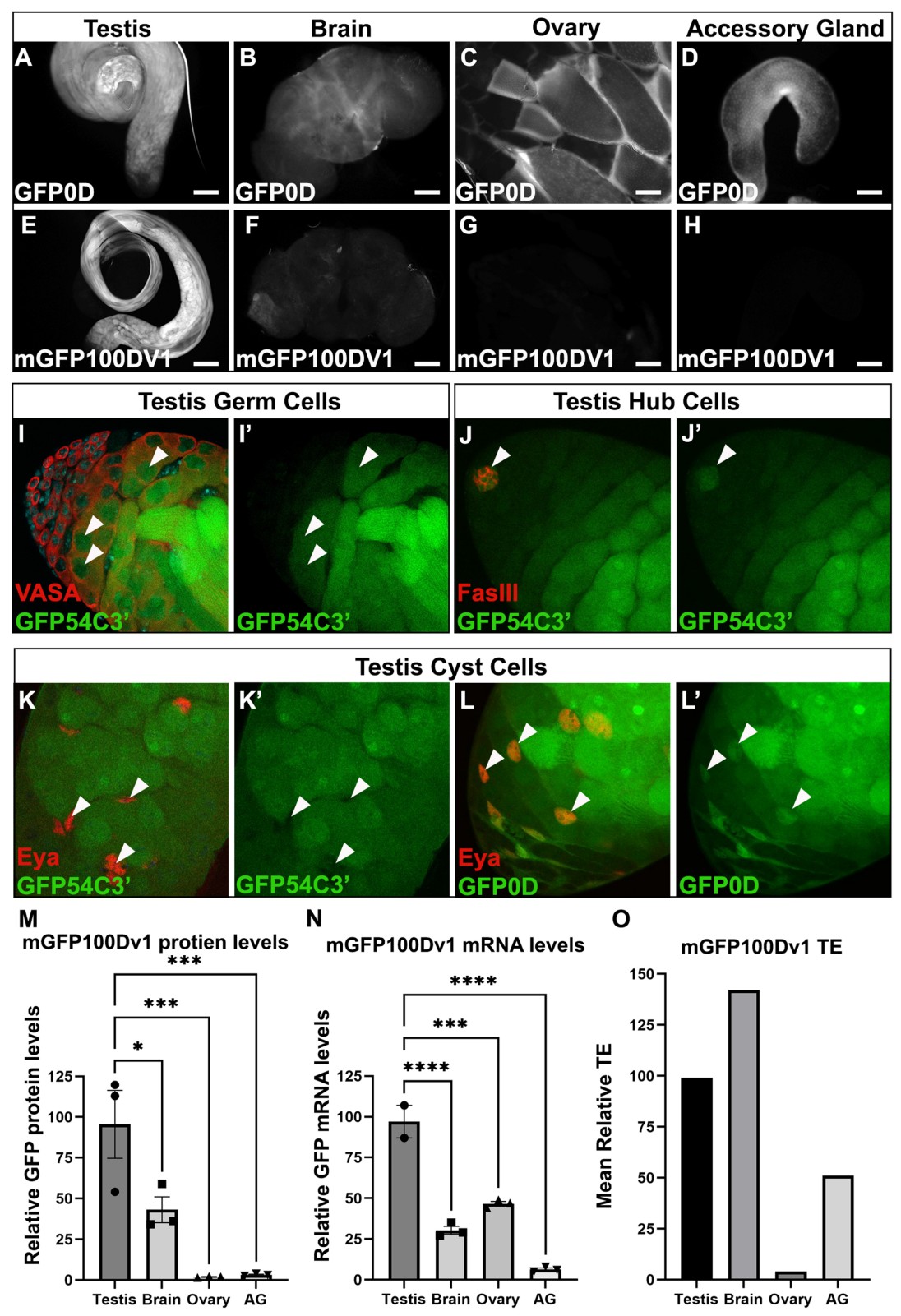

**Figure 3.** The adult testis and brain robustly express rare codon-enriched reporters. (**A–D**) Representative fluorescent images of dissected *GFP0D* adult tissues. Images are taken under identical conditions with fluorescence intensity normalized to testis. (**E–H**) Representative fluorescent images of dissected *mGFP100Dv1* adult tissues. Images are taken under identical conditions with fluorescence intensity normalized to testis. (**I–L′**) Confocal images of adult *GFP54C3′* testis immunostained with antibodies recognizing the indicated cell types. Arrowheads indicate cells of interest, **I**, I′ = germ cells,

*Figure 3 continued on next page*

*Figure 3 continued*

**J, J'**=hub, **K, K'**=somatic cyst cells. Right image in each pair shows the GFP only channel from the left image in the pair. (**L'L'**) Confocal image of adult GFP0D testis immunostained with Eya. Arrowheads = somatic cyst cells. (**L'**) GFP only channel of image in (**L**) (**M**) mGFP100Dv1 protein abundance in dissected adult tissues measured by western blot and plotted as a percentage relative to GFP0D (three replicates, N=10–12 animals each, plotting mean ± SEM, Dunnett's multiple comparison to testis, *p<0.05, ***p≤0.001). See *Figure 3—figure supplement 1* for representative blot image. (**N**) Steady state mRNA levels for heterozygous *mGFP100Dv1/GFP0D* expressing animals measured by Quantitative Real-Time PCR (qRT-PCR). *mGFP100Dv1* mRNA levels are plotted as a percentage relative to GFP0D within each tissue. (2–3 replicates, N=10 animals each, plotting mean ± SEM, Dunnett's multiple comparison to testis, ***p=0.0001,****p<0.0001). (**O**) Translation efficiency of *mGFP100Dv1* in each tissue plotted as a percentage relative to *GFP0D*, plotting mean value. See Methods for details. Scalebars are 100 µm. AG = accessory gland.

The online version of this article includes the following source data and figure supplement(s) for figure 3:

**Figure supplement 1.** Western blot for adult tissues.

**Figure supplement 1—source data 1.** Raw data file and associated PDF figure of uncropped blot presented in *Figure 3—figure supplement 1A* and quantified in *Figure 3—figure supplement 1B* and *Figure 3M*.

**Figure supplement 1—source data 2.** Raw data file and associated PDF figure of uncropped blot presented in *Figure 3—figure supplement 1A* and quantified in *Figure 3—figure supplement 1B* and *Figure 3M*.

**Figure supplement 1—source data 3.** Raw data file and associated PDF figure of uncropped blot quantified in *Figure 3—figure supplement 1B*, and *Figure 3M*.

**Figure supplement 1—source data 4.** Raw data file and associated PDF figure of uncropped blot quantified in *Figure 3—figure supplement 1B*, and *Figure 3M*.

**Figure supplement 2.** The brains of both sexes robustly express rare codon-enriched reporters.

**Figure supplement 3.** Tissue-specific expression is not dependent on the genomic locus of transgene insertion.

**Figure supplement 4.** mRNA length from indicated transgenes.

*3*). Collectively, these results highlight striking differences between *Drosophila* tissues, most notably between the male and female gonad, with regards to rare codon-derived protein expression.

Our findings on rare codon-derived protein expression at a whole tissue level, which are most pronounced in the testis, prompted us to further examine which specific cell types in the testis can robustly express rare codon-enriched reporters. Multiple cell types comprise the *Drosophila* testis, including both somatic and germline cell lineages (*Fairchild et al., 2017*; *Herrera and Bach, 2019*; *Lim et al., 2015*; *Yamashita et al., 2005*). To determine which specific cell types express protein from rare codon-enriched reporters, we used cell type-specific antibodies and assessed co-localization with rare codon-enriched GFP54C3'. Male germ cells (Vasa positive) and the somatic hub cells (FasIII positive), which comprise the male germline stem cell niche, robustly express GFP54C3' (*Figure 3I–J'*), whereas somatic cyst cells that encapsulate the germline cells (Eya positive) do not (*Figure 3K and K'*). Importantly, somatic cyst cells robustly express common codon-enriched GFP0D (*Figure 3L and L'*), indicating that the *ubiquitin* promoter is active in this cell type. Thus, both the male germ cells and the specialized somatic cells of the male germline stem cell niche express protein derived from a rare codon-enriched reporter, while the larger population of somatic cyst cells lack this capability. Together, we find a striking difference between male and female gonads in the expression of protein from rare codon-enriched genes.

## Distinct regulation of rare codon-enriched mRNAs in the testis of flies and humans

Our observed tissue-specific protein expression of rare codon-enriched reporters in the testis and brain could be primarily driven at the protein level, but could also be regulated at the level of mRNA (reviewed in *Radhakrishnan and Green, 2016*). Rare codons can decrease levels of transcription (*Fu et al., 2018*; *Yang et al., 2021*; *Zhao et al., 2021*; *Zhou et al., 2016*) or can negatively impact mRNA stability through a growing number of characterized mechanisms and in numerous model systems (*Bazzini et al., 2016*; *Burow et al., 2018*; *Buschauer et al., 2020*; *Presnyak et al., 2015*; *Radhakrishnan et al., 2016*; *Wu et al., 2019*; *Zhao et al., 2017*). Alternatively, translational repression mechanisms can act independently of mRNA stability, further complicating the relationship between codon optimality and mRNA levels (*Freimer et al., 2018*).

We next assessed whether reporter protein abundance reflects mRNA levels in different tissues. The highest mRNA level for mGFP100Dv1 is found in the testis (*Figure 3N*). Further, mRNA levels of

mGFP100Dv1 and GFP0D are nearly identical in the testis. This result matches our finding at the protein level in the testis (*Figure 3M vs. N*). mGFP100Dv1 mRNA is also detected in several other tissues, but at a reduced level relative to GFP0D. Specifically, we detect mGFP100Dv1 mRNA in the ovary (50% of GFP0D levels), brain (30% of GFP0D), and accessory gland (10% of GFP0D, *Figure 3N*). Among the non-testis tissues examined, only the brain appears able to convert this mRNA into abundant protein. This result is most apparent when plotting the relative translation efficiency (TE) of mGFP100Dv1 in each tissue (*Figure 3O*). In contrast, the ovary is capable of accumulating mGFP100Dv1 mRNA (*Figure 3N*), but translation is impaired (*Figure 3M and O*). Taken together, these results indicate that both mRNA abundance (from the combined effects of transcription/mRNA decay) and translation efficiency of rare codon-enriched reporters are impacted by tissue context. Further, these mRNA results support our finding that, relative to all other tissues examined, gene expression in the testis is less impacted by codon bias.

Mechanistically, each tissue could process the *GFP* reporter pre-mRNA differently, accounting for different protein abundances. To explore this possibility, we assayed for the presence of four possible alternatively spliced m*GFP100Dv1* mRNAs that could be produced from our transgene in each tissue (*Figure 3—figure supplement 4A,B*). We observe no obvious difference in mRNA processing (*Figure 3—figure supplement 4C,D*, *Supplementary file 3*). Hence, mRNA processing does not appear to substantially contribute to tissue-specific differences for our codon-altered reporters. We next sought to put our findings with synthetic reporter mRNA levels and codon content in context with endogenous genes. Many established bioinformatic metrics related to codon usage such as the species-specific tRNA Adaptation Index (stAI), which utilizes tRNA gene copy numbers to predict translation efficiency, reflect variables that do not change within an organism's tissues (*Sabi and Tuller, 2014*). And indeed, we do not observe differences in stAI between *Drosophila* tissues (*Figure 4—figure supplement 1*). Therefore, we developed a modification to the CAI to create a new codon usage metric that reflects codon usage among highly expressed mRNAs that are enriched in a specific tissue. We term this metric tissue-apparent CAI (taCAI). In developing taCAI, we first defined tissue-specific gene sets from publicly available tissue-specific *Drosophila* RNAseq datasets (**Methods**) for each tissue using cutoffs based on relative expression levels (**Methods**). From these tissue-specific gene sets, we then took the 300 most enriched genes for each of 12 adult tissues and obtained the usage frequency for each codon within that set of over-represented genes. The codon usage frequencies for each coding sequence in the genome were then compared against usage frequencies within the tissue-specific gene set to obtain a spread of taCAI values visualized as violin plots. Low taCAI values indicate that mRNAs expressed in a tissue are enriched for rare codons, while high taCAI values indicate that mRNAs expressed in a tissue are enriched for common codons. We calculated taCAI for individual adult tissues in *Drosophila melanogaster* for which RNAseq data exists in FlyAtlas2 (*Krause et al., 2022*; **Methods**).

When visualizing the taCAI distributions for each tissue as a violin plot (*Figure 4A*), we notice a clear tissue-specific trend. The testis and accessory gland, both parts of the male reproductive system, are highly enriched for mRNAs with abundant rare codon usage relative to all other tissues and to the transcriptome as a whole. We note that the accessory gland does not accumulate rare codon-enriched reporters, and the brain does not have a particularly high taCAI (see **Discussion**). In the testis, our computational findings with taCAI match our experimental results using rare codon-enriched *GFP* reporters. Taken together, our results show that among the tissues examined, the *Drosophila* testis accumulates by far the highest mRNA and protein levels for a rare codon-enriched reporter. This finding is consistent with high levels of mRNA for endogenous testis-expressed, rare codon-enriched genes.

We next examined human genes for tissue-specific differences in codon usage. We obtained transcriptomic data from the Human Protein Atlas (HPA; *Uhlén et al., 2015*) project and computed taCAI for each tissue. After correcting for the larger number of distinct human tissues with RNAseq data (37 vs. 12, see **Methods** and *Figure 4—figure supplement 2*), the testis again emerged as a unique rare codon-enriched outlier tissue, along with the pancreas (*Figure 4B*). While available proteomic data for human tissues does not have the same depth as RNAseq data, we took an available human tissue specific proteomic dataset (*Jiang et al., 2020*) and analyzed the CAI of each tissue. In support of our taCAI analysis, again the testis was the top-ranked tissue (meaning it has a high number of proteins derived from mRNAs with abundant rare codons *Figure 4—figure*

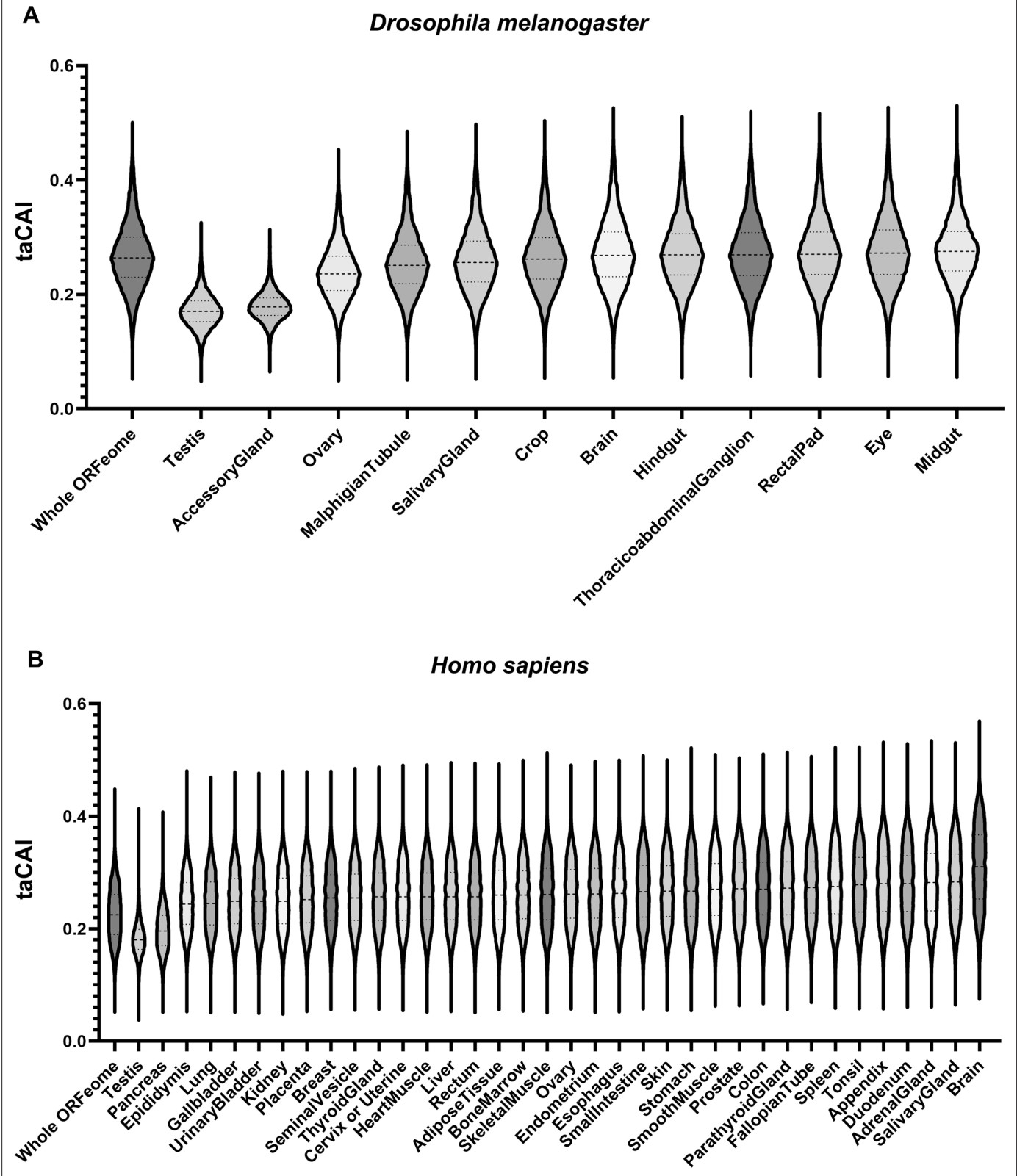

**Figure 4.** Endogenous testis genes are enriched in rare codons. (**A**) issue-apparent CAI (taCAI) for *Drosophila melanogaster* tissues from FlyAtlas2. All tissues analyzed are from adult male animals except the ovary, which is from adult females. Whole ORFeome is plotted as a reference using organismal codon usage frequencies from the Kazusa codon usage database. (**B**) taCAI for human tissues from the Human Protein Atlas. Whole ORFeome is plotted as a reference using organismal codon usage frequencies from the Kazusa codon usage database.

*Figure 4 continued on next page*

*Figure 4 continued*

The online version of this article includes the following figure supplement(s) for figure 4:

**Figure supplement 1.** Pre-existing species-specific tRNA Adaptation Index (stAI) metric does not capture tissue-specific differences in codon usage.

**Figure supplement 2.** Human tissue-apparent CAI (taCAI) using identical cutoffs as fly.

**Figure supplement 3.** Human tissue-specific proteome data indicate that testis-specific proteins are derived from rare codon-enriched genes.

---

*supplement 3*). These data argue for a conserved and distinctive regulation of mRNAs from rare codon-enriched genes in the testis.

## Rare codons in the evolutionarily young gene *RpL10Aa* impact fertility

The abundance of rare codon-enriched mRNAs in the testis suggests that rare codons could limit expression and/or impact function of an mRNA/protein to this tissue. Given this, we next searched for an example where rare codons in a testis-specific gene are critical to the animal. The 'out-of-testis hypothesis' (*Assis and Bachtrog, 2013*; *Kaessmann, 2010*) posits that the testis provides a permissive environment for evolution of new genes. This is based on observations in numerous organisms that young genes tend to be testis-specific and gradually gain broader tissue function as they age (*Betran, 2002*; *Kondo et al., 2017*; *Marques et al., 2005*). The restriction of young genes to the testis has been largely considered a passive consequence of permissive chromatin states during meiosis in the germ cells (*Soumillon et al., 2013*). However, the reduced impact of rare codons on mRNA and protein expression in the testis may provide an alternative mechanism for restricting the expression/function of young genes to this tissue.

To examine the codon content of evolutionarily young genes, we analyzed data on young genes arising from duplication events (*Zhang et al., 2010*). Following a duplication event, there is a parent gene copy that often retains its ancestral function and a child gene copy, which may evolve a new function. We analyzed the codon content of evolutionarily young parent and child gene pairs arising from retrotransposition-mediated duplications identified by *Zhang et al., 2010*. We analyzed the CAI of 96 such parent-child gene pairs. These data reveal that child genes arising from retrotransposition tend to have lower CAI values than their parent genes (*Figure 5A*). We found that in roughly half (44/96) of these retrotransposition events, child genes have maximum expression in the testis among all tissues analyzed (N=12 tissues), while their parent genes were highest expressed in a tissue other than testis (*Supplementary file 4*). These findings imply that rare codons may restrict the expression/function of evolutionarily young genes to the testis.

In several cases, the difference between parent and child can span nearly the entire range of CAI for expressed genes (with the parent at the common end and child at the rare end, *Figure 5A* see *Figure 1T* **for the genomic range**). From these examples, we chose a candidate gene. The gene encoding the testis-specific ribosomal subunit RpL10Aa has the second largest difference in CAI between parent and child gene of any young retrotransposition-mediated duplication in *Drosophila*. The ancestral copy of *RpL10Aa*, *RpL10A*, is a highly conserved member of the 60 s ribosomal subunit. Within the subgenus *Sophophora*, approximately 10 million years ago a gene duplication event, likely mediated by retrotransposition (*Marygold et al., 2007*; *Zhang et al., 2010*), gave rise to two isoforms in the *melanogaster* subgroup (*Figure 5B*). This duplication produced a parent gene copy, *RpL10Ab*, and a child copy *RpL10Aa*. We chose to study *RpL10Aa* because of its young evolutionary age, and because of the striking differences in codon usage and tissue expression pattern between *RpL10Aa* and the parent gene *RpL10Ab*. *RpL10Aa* is highly enriched in rare codons (CAI 0.66) placing it in the rarest 8% of genes organism-wide, is testis-specific in expression pattern (*Figure 5C*), and is very highly expressed in the testis, ranking in the top 2.5% of all genes expressed in the testis. In contrast, *RpL10Ab* has very few rare codons (CAI 0.82), ranking it above 95% of genes organism-wide, and is expressed relatively uniformly throughout the body (*Figure 5C*). That such a drastic reduction in codon bias coupled with high testis-specific gene expression occurred over a short evolutionary period suggests that rare codons may have been positively selected for in *RpL10Aa*.

To examine the role of rare codon enrichment in a young testis gene, we analyzed protein levels for both an endogenous *RpL10Aa* sequence (*RpL10Aa Endo*) and a codon optimized *RpL10Aa* sequence with all common codons (*RpL10Aa Com*). We did this by placing both coding sequences behind the same *ubiquitin* promoter used for our organism-wide GFP reporter screen (*Figure 5D*). To specifically

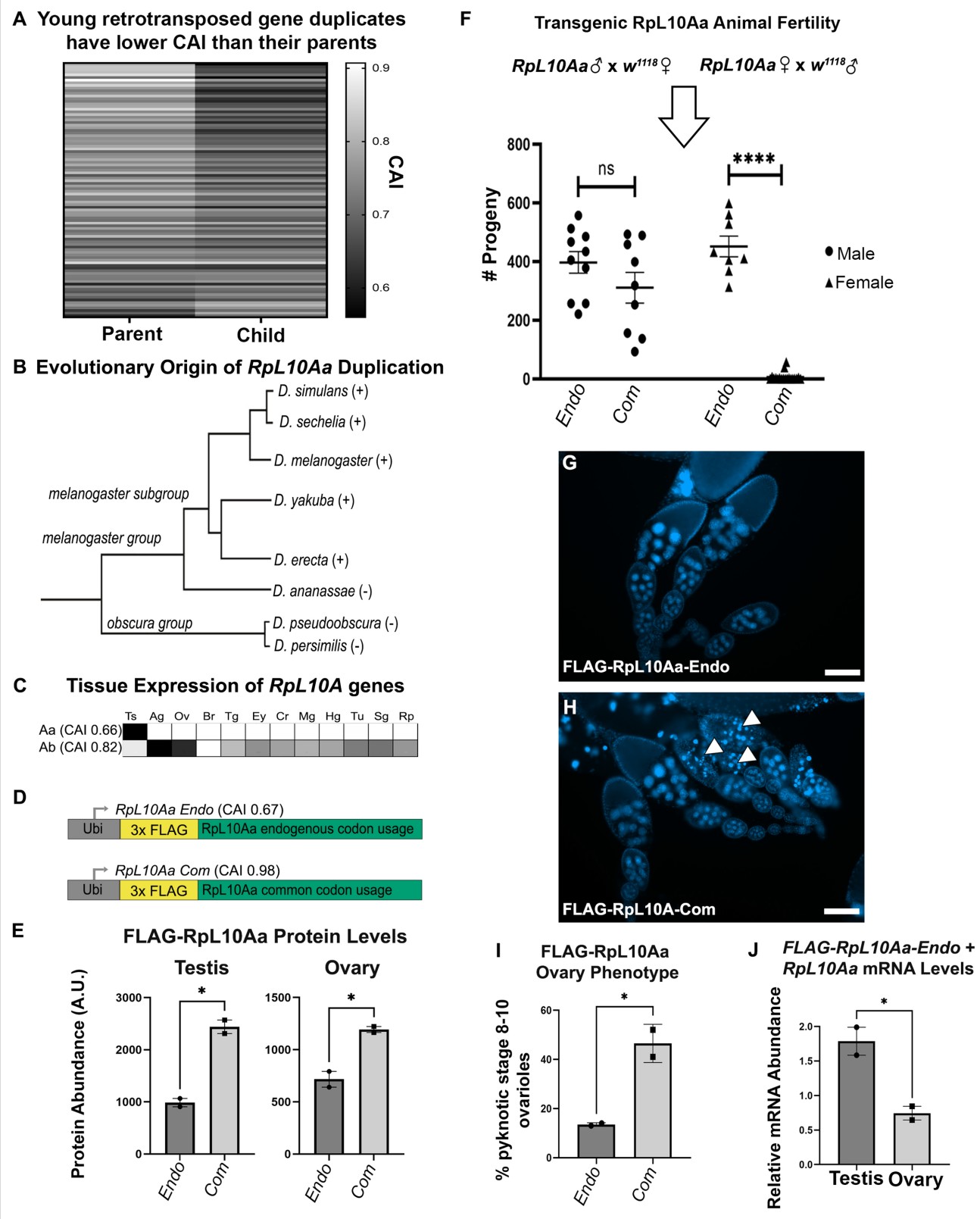

**Figure 5.** Rare codons in RpL10Aa, an evolutionarily young testis gene, impacts fertility. (**A**) Codon usage analysis for young gene duplicate pairs arising from retrotransposition events, identified by *Zhang et al., 2010*. Heatmap represents 96 duplicate gene pairs with the Codon Adaptation Index (CAI) of the parent gene indicated on the left and the CAI of the child gene indicated on the right. (**B**) Phylogenetic tree (*Granzotto et al., 2009*) of the *RpL10Aa* gene duplication event. *RpL10Aa* is only present in the *melanogaster* subgroup as indicated by (+). *RpL10Aa* is not present in *D. ananassae*

*Figure 5 continued on next page*

*Figure 5 continued*

or the *obscura* group as indicated by (−). (**C**) Expression profiles for *RpL10Aa* and *RpL10Ab* based on FlyAtlas2 RNAseq. Darker boxes indicate higher expression. Ts = testis, Ag = accessory gland, Ov = ovary, Br = brain, Tg = thoracicoabdominal gland, Ey = eye, Cr = crop, Mg = midgut, Hg = hindgut, Tu = Malpighian tubule, Sg = salivary gland, Rp = rectal pad. (**D**) Design of *RpL10Aa* transgenes. CAI for transgenic *3xFLAG-RpL10Aa* CDS indicated in parentheses. (**E**) Quantification of transgenic RpL10Aa protein levels in adult testis and young (0–8 hr old) adult ovaries measured by western blot. Protein levels are normalized to total protein stain (two replicates, N=10 animals each, plotting mean ± SEM, unpaired T-test, *p<0.05). Western blot images in *Figure 5—figure supplement 2*. (**F**) *RpL10Aa* fertility assay. Single *RpL10Aa* transgene expressing flies were crossed to single wildtype flies of the opposite sex. Adult progeny resulting from a 10 days mating period were counted. Each point represents progeny from one mating pair. (8–19 replicates per condition, Tukey's multiple comparisons, ns = p>0.05, ****p<0.0001). Legend indicates sex of *RpL10Aa* transgene expressing parent. (**G–H**) Representative fluorescence images of *RpL10Aa* transgene expressing ovaries depicting egg chambers with pyknotic nuclei (arrowheads) at mid-oogenesis in *RpL10Aa Com* animals. Nuclei are stained with DAPI. (**I**) Quantification of pyknotic nuclei phenotype in **G–H**. (**J**) *RpL10Aa Endo* mRNA levels encompassing both transgenic *RpL10Aa Endo* and *RpL10Aa* from the endogenous gene locus. RNA levels are plotted as a ratio relative to codon optimized transgenic *RpL10Aa-Com*.

The online version of this article includes the following source data and figure supplement(s) for figure 5:

**Figure supplement 1.** The *FLAG-RpL10Aa-Endo* reporter transgene is not selectively expressed in the testis.

**Figure supplement 1—source data 1.** Raw data file and associated PDF figure of uncropped blot presented and quantified in *Figure 5—figure supplement 1*.

**Figure supplement 1—source data 2.** Raw data file and associated PDF figure of uncropped blot presented and quantified in *Figure 5—figure supplement 1*.

**Figure supplement 2.** Western blot for RpL10Aa in adult ovary and testis.

**Figure supplement 2—source data 1.** Raw data file and associated PDF figure of uncropped blot presented in *Figure 5—figure supplement 2A* and quantified in *Figure 5E*.

**Figure supplement 2—source data 2.** Raw data file and associated PDF figure of uncropped blot presented in *Figure 5—figure supplement 2A* and quantified in *Figure 5E*.

**Figure supplement 2—source data 3.** Raw data file and associated PDF figure of uncropped blot presented in *Figure 5—figure supplement 2B* and quantified in *Figure 5E*.

**Figure supplement 2—source data 4.** Raw data file and associated PDF figure of uncropped blot presented in *Figure 5—figure supplement 2B* and quantified in *Figure 5E*.

measure RpL10Aa protein from transgenes and not from the endogenous gene locus, we included an N-terminal 3x-FLAG tag in both the endogenous and codon optimized *RpL10Aa* sequences. By western blot, we observe that our ectopically expressed transgenes could be detected in not only the testis, but also the ovary, gut, and brain (*Figure 5—figure supplement 1*). This suggests that the endogenous codon content of *RpL10Aa* alone is not sufficient to restrict gene expression to the testis when driven by a strong ectopic promoter. In comparing RpL10Aa Endo and Com protein levels, we find that RpL10Aa Com protein is 2-fold higher than RpL10Aa Endo protein in both testis and in ovaries (*Figure 5E*, *Figure 5—figure supplement 2*). These measurements establish that codon optimization increases protein abundance from an ectopic *RpL10Aa* gene, both in the testis (where *RpL10Aa* is highly expressed) and in the ovary (where *RpL10Aa* is not normally expressed).

Our previous study of codon-dependent effects of Ras signaling in *Drosophila* identified profound phenotypic differences of a 2-fold, codon-dependent increase in protein expression (*Sawyer et al., 2020*). Indeed, while generating stable lines expressing ectopic *RpL10Aa*, we noticed that *RpL10Aa Com* animals are much less fertile than *RpL10Aa Endo* animals. We hypothesized that female fertility but not male fertility is impacted by codon optimization of the *RpL10Aa* coding sequence, given the tissue specificity of *RpL10Aa* mRNA expression from its endogenous locus (*Figure 5C*). To evaluate male and female fertility for *RpL10Aa Com* and *RpL10Aa Endo* animals, we performed reciprocal single fly crosses between our transgenic flies and wildtype flies. Throughout the course of our fertility assay, we observed high female lethality in *RpL10Aa Com* animals (11/30 females, 1/10 males) and low lethality in *RpL10Aa Endo* animals (2/10 females, 0/10 males). Only crosses where both parents survived the duration of the study were analyzed with respect to fertility. We observe no difference in male fertility between endogenous and codon optimized *RpL10Aa* animals when crossed to wild-type females (*Figure 5F*). For the reciprocal cross (wildtype males crossed to transgenic females), we instead observe drastic differences in female fertility between reporters. *RpL10Aa Com* females produced just 6 viable offspring on average compared with *RpL10Aa Endo* females, which produced on average 452 viable offspring (*Figure 5F*). To more closely examine the cause of female infertility

in *RpL10Aa Com* females, we examined adult ovary morphology. While early (pre-stage 8) ovarioles appear normal, stage 8–10 ovarioles of *RpL10Aa Com* animals contain numerous pyknotic nuclei (*Figure 5G–I*). Therefore, optimizing codon content of *RpL10Aa* in an ectopic reporter context specifically disrupts female (not male) fertility, due to disrupted oogenesis (see **Discussion**).

Our findings suggest that optimizing codons in *RpL10Aa* increases RpL10Aa protein expression in both testis and ovary, but only the ovary is negatively impacted. This could be partly because (as suggested by RNAseq, *Figure 5C*), the testis naturally accumulates (and tolerates) a high level of *RpL10Aa* mRNA, whereas the ovary does not. To assay the relative mRNA abundance of endogenous vs. codon-optimized *RpL10Aa* in our transgenic animals, we measured the relative amount of endogenous and codon-optimized *RpL10Aa* mRNA in both testis and ovary. In these experiments, we used primers that simultaneously detect both possible forms of mRNA from endogenous *RpL10Aa* sequence (both transgenic *RpL10Aa Endo* mRNA and *RpL10Aa* mRNA from the endogenous locus). We separately examined *RpL10Aa Com* mRNA using primers specific to mRNA from this transgene. Whereas the ovary has a lower ratio of endogenous to common *RpL10Aa* mRNA, the testis has the reverse- a higher level of endogenous to common *RpL10Aa* mRNA (*Figure 5J*). We interpret this to reflect the high level of endogenous *RpL10Aa* already present in the testis. Taken together, our findings illustrate an important role for rare codons in tissue-specific gene expression and imply that rare codons in the normally testis-specific gene *RpL10Aa* help to preserve fertility.

## Discussion

Redundancy of the genetic code has long been a mystery of the central dogma of molecular biology. Here, we use *Drosophila melanogaster* to uncover distinct tissue-specific responses to the same genetic code. Our work uncovers fundamental aspects of rare codon biology. We reveal a cliff-like limit on rare codon usage per gene. Near the boundaries of this limit, however, we demonstrate that individual tissues have different tolerances for protein expression from genes enriched in rare codons. Tissues with a high tolerance for rare codons include the brain of both sexes and the testis, but not the ovary. Taking a closer look at the testis, we find the male germ cells and somatic cells of the germline stem cell niche of the testis robustly express protein from rare codon-enriched reporters, while the somatic cyst cells do not. In developing a new metric for tissue-specific codon usage, taCAI, we reveal endogenous genes in the testis of both *Drosophila* and humans show an abundance of rare codon-enriched mRNAs, suggesting that rare codon-enriched genes play an essential and conserved role in testis biology. In search of a physiologically relevant role for rare codon tolerance in testis biology, we demonstrate that endogenously rare codons may restrict the expression and function of the evolutionarily young gene *RpL10Aa* to the testis. Codon optimization of an *RpL10Aa* transgene disrupts female fertility and organism viability. Taken together, our results uncover clear tissue-specific differences in the impact of codon usage bias and partially support a novel role for rare codons in restricting the expression of evolutionarily young genes to the testis.

### Codon bias as an important parameter for organism-wide and tissue-specific mRNA and protein expression

Here, our sensitive reporter-based approach suggests a strong correlation between CAI and reporter GFP abundance in whole *Drosophila* ($R^2$=0.82, p<0.0001). We note that previous data from bioinformatic approaches in human tissues are conflicting regarding correlations between CAI and tissue-specific mRNA and protein expression (*Plotkin et al., 2004*; *Sémon et al., 2006*). Further, similar bioinformatic studies suggested only weak codon-dependent differences between tissues in *Drosophila* (*Payne et al., 2019*). The sensitivity of our reporter and the ability of our expression platform to rapidly test various sequences over a narrow CAI range proved helpful in defining rare codon limits for protein expression, which closely mirror existing mRNA expression data for the testis. Our study reveals a strict limit to rare codon usage on a genomic scale. This limit could reflect the tipping point in rare codon usage before competition between transcription and mRNA decay favors degradation over translation (*Buschauer et al., 2020*; *Radhakrishnan et al., 2016*).

We also observe clear differences in the impact of codon usage bias between *Drosophila* tissues. Importantly, we show how distinct tissues respond differently to the same genetic code. It is well appreciated that tissue-specific chromatin environments play critical roles in transcriptional regulation. Our findings here suggest that tissue-specific tolerances for rare codons play a similar role at the levels

of translation and transcription/mRNA decay. There are several likely candidate mechanisms underlying tissue-specific impacts of codon bias, including differences in levels of RNA decay machinery or tRNAs between tissues. Several RNA decay pathways have been linked to rare codon usage and would be of interest to follow up on in our system (*Buschauer et al., 2020*; *Radhakrishnan et al., 2016*). Levels of expressed tRNAs have been previously measured in *Drosophila* at the organismal level (*Moriyama and Powell, 1997*; *White and Tener, 1973*), but not at the level of individual tissues. Quantifying tRNA expression is not the same as measuring tRNA activity, however, as tRNAs exhibit rigid folding structure, distinct amino acid carrying status when active, and require post transcriptional modifications. The contribution of tRNA regulation to tissue-specific codon bias regulation can be explored further in the future.

We note that mechanisms, which lead to rare codon mRNA/protein accumulation, may be distinct in different tissues. For example, our taCAI analysis of RNAseq data revealed that the testis but not the brain is enriched for mRNAs with abundant rare codons. Recent work by other groups, however, demonstrates that rare codons have less influence on mRNA stability in central nervous system compared to whole embryo (*Burow et al., 2018*), supporting our reporter-based finding here that the brain has unique biology pertaining to rare codons. In addition, we find that while the testis is capable of translating *GFP0D* and *mGFP100Dv1* mRNA with relatively equal efficiency, the brain appears to translate *mGFP100Dv1* even more efficiently than *GFP0D*. This suggests that the brain may have specialized translational machinery or tRNA pools better adapted to rare codons than to common ones, whereas the testis is equally adapted to both rare and common codon pools. Further, while the accessory gland did not robustly express our rare codon-enriched reporters, our taCAI analysis suggests this male reproductive tissue is also enriched in mRNAs with abundant rare codons. Now that we have pinpointed interesting differences between the male reproductive tract, brain, and other tissues, these differences can be exploited in future studies to reveal the underlying molecular mechanisms.

## Rare codons and gene evolution: a potential role for the testis

Here, we find a conserved enrichment for rare codons in testis-specific genes. Further, we demonstrate a role for rare codons in the evolutionarily young *RpL10Aa*, a known testis-specific gene. Optimizing the codon sequence of this testis-specific gene causes ectopic high expression in the ovary and results in female infertility. Specifically, the ovary is known to activate a mid-oogenesis apoptotic checkpoint (*Pritchett et al., 2009*), and our observed phenotypes are consistent with ectopic codon-optimized *RpL10Aa* disrupting mid-oogenesis. We note that our *ubi-* promoter is stronger in the soma than the germline (*Figure 3—figure supplement 1C*), which may suggest a somatic origin of the phenotypes we observe. We speculate that ectopic RpL10Aa protein in the ovary may act as a dominant negative relative to the function of the parent gene *RpL10Ab* (64% identity between these proteins), which is required for female fertility (*Wonglapsuwan et al., 2011*). However, an alternative is that the phenotypes we observe are mRNA mediated, rather than protein mediated.

Tissue specificity imparted by rare codons may have functional relevance to the 'out-of-testis' hypothesis for new gene evolution (*Kaessmann, 2010*). This hypothesis proposes that the testis acts as a 'gene nursery' to promote evolution and neofunctionalization of young genes through its permissive expression environment and intense selective pressures from both mate and sperm competition. This is based on observations that testes are the fastest evolving tissue and that young genes are often restricted in expression to the testis and gain broader expression patterns as they age.

While the current model is that permissive chromatin states passively allow expression of young genes resulting from retrotransposition events specifically in the testis (*Kaessmann, 2010*). Here, we suggest that rare codon content may also fulfill this role and could be positively selected for an evolutionarily young genes in *Drosophila*. Male-biased genes have also been characterized to have rarer codon usage than female-biased genes and non-sex-biased genes in numerous plant species (*Darolti et al., 2018*; *Song et al., 2017*; *Whittle et al., 2007*) and *Drosophila* (*Hambuch and Parsch, 2005*), and have rarer codon usage than non-sex-biased genes in zebrafish and stickleback (*Yang et al., 2016*). Thus, our findings may be widely applicable to sexually reproducing species across phylogenetic kingdoms. Taken together, we highlight codon usage as a critical and under-appreciated aspect of tissue-specific gene regulation that merits greater attention, notably in the fields of evolutionary and developmental biology.

# Materials and methods

## Key resources table

| Reagent type (species) or resource | Designation | Source or reference | Identifiers | Additional information |
|---|---|---|---|---|
| antibody | anti-GFP (rabbit polyclonal) | Invitrogen | A11122 | (1:1000) |
| antibody | anti-FLAG M2 (mouse monoclonal) | Sigma | F1804 | (1:500) |
| antibody | IRDye 800CW Donkey anti-rabbit IgG Secondary Antibody (clonality unspecified) | LI-COR | 926–32213 | (1:10,000) |
| antibody | IRDye 800CW Donkey anti-mouse IgG Secondary Antibody (clonality unspecified) | LI-COR | 926–32212 | (1:10,000) |
| recombinant DNA reagent | pBID-Ubi | this paper | | modified from Addgene Plasmid #35,200 |
| gene (D. melanogaster) | RpL10Aa | NA | FBgn0038281 | |
| genetic reagent (D. melanogaster) | attP40 | Model System Injections | | y1() M{RFP[3xP3.PB] GFP[E.3xP3]=vas int.Dm}ZH2A w[*]; P{y[+t7.7]=CaryP}attP40 |
| genetic reagent (D. melanogaster) | w$^{1118}$ | BDSC | #3605 | |
| genetic reagent (D. melanogaster) | Ubi-GFP0D | *Sawyer et al., 2020* | | attP40 (2 L) |
| genetic reagent (D. melanogaster) | Ubi-GFP30D | this paper | | attP40 (2 L) |
| genetic reagent (D. melanogaster) | Ubi-GFP50D | *Sawyer et al., 2020* | | attP40 (2 L) |
| genetic reagent (D. melanogaster) | Ubi-GFP60Dv1 | this paper | | attP40 (2 L) |
| genetic reagent (D. melanogaster) | Ubi-GFP60Dv2 | this paper | | attP40 (2 L) |
| genetic reagent (D. melanogaster) | Ubi-GFP60Dv3 | this paper | | attP40 (2 L) |
| genetic reagent (D. melanogaster) | Ubi-GFP70D | this paper | | attP40 (2 L) |
| genetic reagent (D. melanogaster) | Ubi-GFP80D | this paper | | attP40 (2 L) |
| genetic reagent (D. melanogaster) | Ubi-GFP90D | this paper | | attP40 (2 L) |
| genetic reagent (D. melanogaster) | Ubi-GFP100D | this paper | | attP40 (2 L) |
| genetic reagent (D. melanogaster) | Ubi-GFP50C3′ | this paper | | attP40 (2 L) |
| genetic reagent (D. melanogaster) | Ubi-GFP60C3′ | this paper | | attP40 (2 L) |
| genetic reagent (D. melanogaster) | Ubi-GFP70C3′ | this paper | | attP40 (2 L) |
| genetic reagent (D. melanogaster) | Ubi-GFP80C3′ | this paper | | attP40 (2 L) |
| genetic reagent (D. melanogaster) | Ubi-GFP90C3′ | this paper | | attP40 (2 L) |
| genetic reagent (D. melanogaster) | Ubi-GFP50C5′ | this paper | | attP40 (2 L) |
| genetic reagent (D. melanogaster) | Ubi-GFP54C3′ | this paper | | attP40 (2 L) |
| genetic reagent (D. melanogaster) | Ubi-mGFP100Dv1 | this paper | | attP40 (2 L) |

*Continued on next page*

*Continued*

| Reagent type (species) or resource | Designation | Source or reference | Identifiers | Additional information |
|---|---|---|---|---|
| genetic reagent (*D. melanogaster*) | *Ubi-mGFP100Dv2* | this paper | | *attP40 (2 L)* |
| genetic reagent (*D. melanogaster*) | *Ubi-mGFP100Dv3* | this paper | | *attP40 (2 L)* |
| genetic reagent (*D. melanogaster*) | *Ubi-mGFP100Dv4* | this paper | | *attP40 (2 L)* |
| genetic reagent (*D. melanogaster*) | *Ubi-mGFP100Dv5* | this paper | | *attP40 (2 L)* |
| genetic reagent (*D. melanogaster*) | *Ubi-mGFP100Dv6* | this paper | | *attP40 (2 L)* |
| genetic reagent (*D. melanogaster*) | *Ubi-mGFP100Dv7* | this paper | | *attP40 (2 L)* |
| genetic reagent (*D. melanogaster*) | *Ubi-mGFP100Dv8* | this paper | | *attP40 (2 L), attP2 (3 L)* |
| genetic reagent (*D. melanogaster*) | *Ubi-3xFLAG-RpL10Aa Endo* | this paper | | *attP40 (2 L)* |
| genetic reagent (*D. melanogaster*) | *Ubi-3xFLAG-RpL10Aa Com* | this paper | | *attP40 (2 L)* |

## Generation of codon-modified reporters in *Drosophila*

Codon-modified exon sequences for *GFP* and *RpL10Aa* were designed according to the codon usage frequencies in the *Drosophila melanogaster* genome taken from the Kazusa codon usage database (https://www.kazusa.or.jp/codon/). The *mCherry* sequence used in generating fusion proteins was obtained from NovoPro pADH77. Sequences were subsequently generated through gene synthesis (ThermoFisher Scientific and Invitrogen, Waltham, MA) and cloned into a pBID-Ubi plasmid (modified from Addgene Plasmid #35200) or were directly synthesized into the desired pBID-Ubi plasmid cloning site (Twist Bioscience, South San Francisco, CA). Plasmids were grown in NEB 5-alpha competent *E. coli* cells (NEB #C2987), transformed according to the manufacturer's protocol, and purified with a ZymoPure II Plasmid Midipred Kit (Zymo Research). Plasmids were injected into either *attP40 (2 L)* or *attP2 (3 L)* flies by Model System Injections (Durham, NC). Full sequences for the synthesized genes are provided in *Supplementary file 1*. Sequences for the *Ubi-p63E* promoter and transgenic reporter UTRs are provided in *Supplementary file 3*.

## Fly stocks

All flies were raised at 25 °C on standard media unless noted otherwise (Archon Scientific, Durham, NC). The following stock was obtained from the Bloomington *Drosophila* Resource Center $w^{1118}$ (#3605). The following stocks were generated for this study: GFP30D, GFP60Dv1, GFP60Dv2, GFP60Dv3, GFP70D, GFP80D, GFP90D, GFP100D, GFP50C5', GFP50C3', GFP60C3', GFP70C3', GFP80C3', GFP90C3', GFP54C3', mGFP100Dv1, mGFP100Dv2, mGFP100Dv3, mGFP100Dv4, mGFP100Dv5, mGFP100Dv6, mGFP100Dv7, mGFP100Dv8, RpL10Aa Endo, RpL10Aa Com.

## Protein quantifications

Protein samples were prepared as in *Sawyer et al., 2020*. Briefly, tissues or whole animals were homogenized in Laemmli buffer (10 tissues in 50 µl or five animals in 100 µl) on ice, then boiled for 5 min. Samples were separated on 12% sodium dodecyl sulfate polyacrylamide gels by electrophoresis (SDS-PAGE) at 200 V. Proteins were transferred onto nitrocellulose membranes using an iBlot 2 Dry Blotting System (Invitrogen, Waltham, MA) set to 20 V, 6 min. Total protein was quantified using Revert700 Total Protein Stain Kit (LI-COR Biosciences, Lincoln, NE) according to the manufacturer's protocol for single channel imaging. For normalization, we note that total protein stain was superior to antibodies against single housekeeping genes (Tubulin) due to variable expression of these proteins between tissues. The following antibodies were used: rabbit anti-GFP (1:1000, Invitrogen, cat#A11122), anti-FLAG M2 (1:500, Sigma, St. Louis, MO, cat#F1804), IRDye 800CW (1:10,000,

LI-COR Biosciences, anti-rabbit or anti-mouse). Signal was detected using a LI-COR Odyssey CLx and analyzed using Image Studio version 5.2 (LI-COR Biosciences). Original source data for all western blots have been provided.

## CAI

CAI calculations for transgenic reporters and endogenous *Drosophila* genes were performed using CAIcal (*Puigbò et al., 2008*). For CAI calculation of endogenous genes, the standalone version of CAIcal was used with input CDS sequences from the *Drosophila melanogaster* genome version r 6.22, and the *Drosophila melanogaster* codon usage table from the Kazusa codon usage database (*Nakamura et al., 2000*).

## taCAI

Tissue-specific RNA sequencing data was obtained for *Drosophila* from FlyAtlas2 and for humans from the Human Protein Atlas. We defined genes as being tissue-specific in *Drosophila* if they were highly expressed in the tissue of interest, and not highly expressed in any other tissue. Specifically, genes had to be among the top 40% of genes expressed in the tissue of interest and excluded from the top 25% of genes expressed in any other tissue analyzed. We chose a thresholding, rather than a fold change, approach to determine tissue specificity. This was because some tissues have a much higher raw expression value than others (the median expression value is more than double for one tissue vs. another tissue in some cases), which confounds fold change approaches. We chose to analyze the 300 most enriched tissue-specific genes per tissue because any value higher than 300 prevented us from examining an equal number of genes in each tissue. Applying the same gene selection filter to human tissues revealed that testis genes were most enriched in rare codons among the 37 tissues analyzed (*Figure 4—figure supplement 1*), however, the 3-fold larger number of tissues in humans compared to flies (37 vs. 12) prompted us to explore additional, more stringent cutoffs of tissue specificity for the human data. Towards this end, we defined genes as being tissue-specific in human if they were highest expressed in the tissue of interest and had a TPM <1 in all other tissues analyzed (*Figure 4B*). Codon usage frequency was calculated using the Kazusa Countcodon program (https://www.kazusa.or.jp/codon/countcodon.html). We used the resulting tissue-specific codon frequency table to compute taCAI values for each tissue using the following equation:

$$taCAI_t = 1 - \sum_{i=1}^{N} CAI_i$$

where $N$ is the number of genes in the genome and $CAI_i$ is the codon adaptation index value calculated for gene    using the tissue-specific codon frequency table for tissue $t$. This calculation returns a genome-sized distribution of values, where the codon usage of each gene is evaluated relative to the codon usage of a pre-defined tissue-specific gene set. The distribution of taCAI values for a given tissue can be visualized as a violin plot and these distributions can be compared between tissues. To additionally visualize how the codon usage of each tissue compares to the codon usage of the entire genome, we computed the genomic taCAI distribution using the following equation:

$$taCAI_g = 1 - \sum_{i=1}^{N} CAI_i$$

where $N$ is the number of genes in the genome and $CAI_i$ is the codon adaptation index value calculated for gene    using the codon frequency table for the entire genome $g$.

## stAI

Species-specific tRNA Adaptation Index (stAI) values were calculated for the same *Drosophila* tissue-specific gene sets used for calculating tissue-apparent CAI (taCAI). This was done using the standalone stAIcalc program (*Sabi et al., 2010*) with input tRNA gene copy numbers from the genomic tRNA database (http://gtrnadb.ucsc.edu/index.html, *Drosophila melanogaster* Aug. 2014 BDGP Release 6 /dm6). The standalone stAIcalc application allows for a user defined variable hill climb stringency setting. stAI calculations were performed using both the minimum allowed hill climb stringency setting and the maximum allowed hill climb stringency setting.

## qRT-PCR

For tissue-specific measurements of *mGFP100Dv1* mRNA, animals heterozygous for both m*GFP100Dv1* and *GFP0D* were aged 3–7 days at 25 °C on standard fly medium supplemented with wet yeast (Archon Scientific, Durham, NC). For tissue-specific measurements of *RpL10Aa-Endo* mRNA, animals heterozygous for both *RpL10Aa-Endo* and *RpL10Aa-Com* were aged 3–7 days at 25 °C on standard fly medium supplemented with wet yeast. For whole larval measurements of GFP reporter mRNA, WL3 larvae heterozygous for the indicated reporter and *GFP0D* were raised at 25 °C on standard fly medium. Where applicable, dissections were performed in RNase-free phosphate buffered saline and completed in under 2 hr. After dissection, tissues were immediately homogenized in TRIzol reagent (N=10 animals per tissue per replicate, 500 µl TRIzol reagent) and snap frozen in liquid nitrogen before storage at –80 °C. For whole larval samples, five WL3 larvae were homogenized in 500 µl TRIzol reagent. RNA was purified according to the manufacturer's protocol, using glycogen as a carrier and resuspending in molecular grade water. RNA was then treated with DNase I at room temp for 15 min before terminating the reaction by adding 2.5 mM EDTA and incubating at 65 °C for 10 min then storing at –80 °C. Quantification of RNA was performed on either a NanoDrop spectrophotometer or Qubit 3 fluorometer and samples were diluted to match the concentration of the lowest concentration sample. Equal amounts of RNA for all samples directly compared to one another were simultaneously transcribed into cDNA using iScript cDNA synthesis kit (BIO-RAD, Hercules, CA, cat#170–8891) according to the manufacturer's protocol within 7 days of the initial dissections to preserve sample quality. No Reverse Transcriptase (NRT) controls were also run simultaneously for each sample to control for genomic DNA contamination. Quantitative Real-Time PCR (qRT-PCR) was run simultaneously on all samples compared to one another, corresponding NRT controls, and No Template Controls (NTC) for each primer pair using Luna Universal qCPR Master Mix (NEB, Ipswich, MA, #M3003) following the manufacturer's protocol (1 µl cDNA per 10 µl reaction). A CFX384 Touch Real-Time PCR Detection System (BIO-RAD) was used for cDNA amplification and detection of FAM/SYBR Green fluorescence. qRT-PCR run data was analyzed using BIO-RAD CFX Manager software. Primer sequences are provided in *Supplementary file 5*. Primer binding efficiencies were analyzed over a wide range of genomic DNA template concentrations. We note that primers against 0D, 30D, 50D, 60Dv1, 70D, 100D, 50C5', and mGFP100Dv1 had near identical binding efficiencies, whereas primers against 60Dv2, 60Dv3, 80D, 90D, 50C3', 60C3', 70C3', 80C3', and 90C3' had slightly higher binding efficiencies. Amplification curves for all samples had characteristic S-shapes and each qPCR primer pair yielded a single melting peak. All cDNA samples reached threshold several cycles earlier than NRT and NTC controls, except whole larval 80D, which was within one cycle of the detection limit. For detection of tissue-specific *mGFP100Dv1* mRNA levels, three technical replicates for all samples yielded very consistent CT values. A single testis sample, however, yielded an outlier delta CT value (*mGFP100Dv1* minus *GFP0D*, Tukey's multiple comparisons, p<0.0001), and was removed from the analysis after bleach-agarose electrophoresis revealed that this sample lacked detectable RNA. For tissue-specific detection of *RpL10Aa-Endo* and *RpL10Aa-Com*, and for whole larvae detection of GFP reporters, two technical replicates yielded consistent CT values. Relative transcript abundance was calculated using the $2^{\Delta CT}$ method to internally normalize expression of the indicated GFP reporters or *RpL10Aa-Endo* against the expression of the respective codon optimized transgene. Graphs were generated using GraphPad Prism 9.2.0.

## Translation efficiency calculations

Mean relative translation efficiency was calculated by dividing the average protein abundance (normalized to GFP0D) by the average mRNA abundance (normalized to GFP0D) and converting to a percentage. Only average values were used in the calculation, because protein and mRNA data were not obtained from paired samples.

## Fluorescence imaging

Whole larval screening and image acquisition was on either a Zeiss SteREO Discovery.V12 (Zeiss Achromat S 0.63 x FWD 107 mm objective) and Zeiss AxioCam ICc 5 camera (images in *Figure 1*), or on a Leica MZ10 F stereoscope (Leica Plan APO 1.0 x objective #10450028) and Zeiss AxioCam MRc r2.1 camera (images in *Figure 2*). For larval screening, we assayed live larvae (a minimum of 50 larvae per line were examined in the screen). To immobilize larvae for imaging, WL3 larvae were

anesthetized using di-ethyl ether for 3.5 min as described in *Kakanj et al., 2020*. Larval and adult tissues were prepared for imaging by dissecting in 1 x PBS, fixing in 1 x PBS, 3.7% paraformaldehyde, 0.3% Triton-X for 30 min, and staining with Hoechst 33,342 (1:1500, Life Technologies, Carlsbad, CA, #c10339). Tissues were mounted in Vectashield (Vector Laboratories Inc, Burlingame, CA). Tissue images were acquired on an upright Zeiss AxioImager M.2 without Apotome processing and with the Apotome unit positioned out of the light path (Zeiss 10 x NA 0.3 EC Plan-Neofluar objective) using a Zeiss Axiocam 503 mono version 1.1. Imaging conditions were identical between tissues to be visually compared (noted in the figure legends). Image processing was performed in FIJI (formatting) and Photoshop (applying levels adjustment layer uniformly across panels to be visually compared).

## 3' RACE

3' Rapid Amplification of cDNA Ends (3' RACE) was performed using the Roche 5'/3' RACE kit, $2^{nd}$ generation (Roche Holding AG, Basel, Switzerland, cat# 03353621001) following the manufacturer's protocol. RNA from whole WL3 larvae expressing *GFP0D* was used as a template. The template-specific SP5 primer sequence used was 5'-CCACAAGCTGGAGTACAACTACAACAGC-3'. Products from the 3' RACE reaction were sequenced by Eurofins Genomics (Eurofins Genomics, Louisville, KY) using the following primer 5'-CGATAACCACTACCTGAGCACCC-3'.

## Acknowledgements

The following kindly provided reagents used in this study: Bloomington *Drosophila* Stock Center, Developmental Studies Hybridoma Bank. David MacAlpine (Duke) provided valuable technical guidance for RT-PCR. Christopher Nicchitta (Duke) and William Marzluff (UNC Chapel Hill) provided valuable technical guidance for determining transgenic UTRs. Braden Tierney (Cornell) provided technical support in downloading FlyAtlas2 data. We thank Daniel Lew and Zhao Zhang (Duke) for comments on the manuscript. This project was supported by ACS grant RSG-128945 to DF, an NSF GRFP grant to SA, NIH grants R01CA94184 and P01CA203657 to CMC, and NIHG grants R35 GM140844 and R01 HL111527 to AL.

## Additional information

### Funding

| Funder | Grant reference number | Author |
|---|---|---|
| American Cancer Society | RSG-128945 | Donald T Fox |
| National Science Foundation | GRFP | Scott R Allen |
| National Institutes of Health | R01CA94184 | Christopher M Counter |
| National Institutes of Health | P01CA203657 | Christopher M Counter |
| National Institutes of Health | R35GM140844 | Alain Laederach |
| National Institutes of Health | R01HL111527 | Alain Laederach |

The funders had no role in study design, data collection and interpretation, or the decision to submit the work for publication.

### Author contributions

Scott R Allen, Conceptualization, Data curation, Formal analysis, Funding acquisition, Investigation, Methodology, Project administration, Resources, Software, Supervision, Validation, Visualization, Writing - original draft, Writing - review and editing; Rebeccah K Stewart, Data curation, Formal analysis, Investigation, Methodology, Validation, Visualization, Writing - review and editing; Michael Rogers, Data curation, Formal analysis, Investigation, Methodology, Resources, Validation,

Visualization, Writing - review and editing; Ivan Jimenez Ruiz, Data curation, Formal analysis, Investigation, Methodology, Resources, Software, Validation; Erez Cohen, Data curation, Formal analysis, Investigation, Methodology, Resources, Software, Validation, Writing - review and editing; Alain Laederach, Conceptualization, Funding acquisition, Investigation, Project administration, Writing - review and editing; Christopher M Counter, Conceptualization, Funding acquisition, Supervision; Jessica K Sawyer, Conceptualization, Investigation, Project administration, Supervision, Writing - review and editing; Donald T Fox, Conceptualization, Formal analysis, Funding acquisition, Investigation, Project administration, Supervision, Visualization, Writing - original draft, Writing - review and editing

#### Author ORCIDs
Scott R Allen http://orcid.org/0000-0002-4809-0493
Alain Laederach http://orcid.org/0000-0002-5088-9907
Christopher M Counter http://orcid.org/0000-0003-0748-3079
Donald T Fox http://orcid.org/0000-0002-0436-179X

#### Decision letter and Author response
Decision letter https://doi.org/10.7554/eLife.76893.sa1
Author response https://doi.org/10.7554/eLife.76893.sa2

## Additional files

#### Supplementary files
- Supplementary file 1. Full coding sequences of reporters.
- Supplementary file 2. Occurrences of codons known to impair translation do not explain reporter expression patterns.
- Supplementary file 3. Sequences of pBID-Ubi plasmid promoter and UTRs.
- Supplementary file 4. Identity of child genes that gained max expression in the testis.
- Supplementary file 5. Primer sequences for qRT-PCR and RT-PCR experiments.
- Transparent reporting form

#### Data availability
All data generated or analysed during this study are included in the manuscript and supporting file; Source Data files have been provided for Figs 1, 3, and 5.

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
