## [Editor Report]

This report is significant in providing strong evidence that differences in codon optimality in mRNAs can underlie tissue-specific differences in expression in fruit flies and that this mechanism is at play in restricting expression of an evolutionarily young ribosomal protein gene with higher than average rare codon usage to the testis versus ovaries in a manner critical for female fertility. The work breaks new ground in identifying codon usage as a basis for tissue-specific gene expression in animals.

---

## [Decision Letter]

**Decision letter after peer review:**

Thank you for submitting your article "Distinct responses to rare codons in select *Drosophila* tissues" for consideration by *eLife*. Your article has been reviewed by 3 peer reviewers, one of whom is a member of our Board of Reviewing Editors, and the evaluation has been overseen by James Manley as the Senior Editor. The following individual involved in review of your submission has agreed to reveal their identity: Yi Liu (Reviewer #2).

Essential revisions:

All three reviewers agreed that the reported results are significant in providing evidence that differences in codon optimality in mRNAs can underlie tissue-specific differences in expression and that this phenomenon operates in restricting expression of an evolutionarily young endogenous gene to the testis versus ovaries in a manner important for female fertility in the fruit fly. At the same time, each noted that the underlying molecular mechanisms have not been probed, and that the scientific quality of the work would be enhanced considerably by additional experiments to determine whether the differences in expression arise primarily at the translational level or also/rather involve altered mRNA turnover or transcription in response to poor codon usage outside of testis and brain. To this end, the authors are requested to (i) provide measurements of reporter mRNA abundance in parallel with their measurements of reporter protein expression throughout the work; (ii) establish that the lengths of the mGFP100DV1 and RPL10a_endo reporter transcripts do not vary between testis and ovary; (iii) verify that the Rpl10A_endo reporter is expressed selectively in testis by examining its expression in other tissues beside testis and ovaries. In addition to performing these additional experiments/analyses, the authors should address all of the other comments of the referees by adding the appropriate additional analyses or making the necessary revisions of text or figures. In those instances where a follow-up experiment has been proposed (in addition to those listed above), but you elect not to conduct it, it may be necessary for you to modify the text to acknowledge that its absence is a shortcoming of the current work."

*Reviewer #1 (Recommendations for the authors):*

– Perhaps the authors would consider inverting the scale for the new taCAI metric so that the values would be lower than average, rather than higher than average, for tissues where the most abundant mRNAs tend to have low versus high CAI values.

– It's unclear from the experiments presented whether it is the number of rare codons or the proportion of rare codons in an mRNA that dictates the repression of expression above the 50% threshold in their reporters. It seems easier to imagine that it is the number of rare codons encountered during translation of an mRNA that contribute cumulatively to translational repression/mRNA decay. If so, short mRNAs with a high percentage of rare codons might be expressed above the 50% threshold because they can be completely translated without encountering the threshold number of rare codons. Can they address this point bioinformatically by examining native mRNAs, eg. to recalculate Figure 1S for short vs long mRNAs? Are there any studies in the literature that would allow the authors to comment on this issue in the Discussion?

– If the protein expression data in Figure 3G were normalized for the mRNA levels in Figure 3N to obtain a measure of translation efficiency (TE), they could conclude that the TEs of the reporter are dramatically reduced in the ovary and AG, but comparable in the testis and brain. This in turn would suggest that impaired translation predominates, and increased mRNA turnover merely contributes, to the failure to express reporters exceeding 50% rare codons. Perhaps this would be a useful interpretation to consider adding to the paper.

– Could the authors try to improve their presentation of the analysis in Figure 5A to make it easier to follow what they have done, and what they conclude from the difference in the distributions of data points in the 2nd and 3rd columns, and whether this difference is statistically significant.

– It would be worth comparing the reporter mRNA levels to complement the protein expression data in Figure 5E to determine whether the increased expression of the Com reporter in ovaries occurs primarily at the level of translation with minimal contribution from increased mRNA stability.

*Reviewer #2 (Recommendations for the authors):*

1. Figure 2. Did the authors try to check the tissue-specific expression of the series of reporters used Figure 1? One would predict that some of reporters should also show testis/brain-specific expression.

2. Figure 2N. The Y axis labeling is hard to understand. Suggest to change to "Relative GFP mRNA level". Are the mRNA differences between the two reporters in different tissues due to different mRNA decay rates? In addition to Burrow 2018 study, Zhao et al., (2017 NAR) also demonstrate an impact of codon usage on mRNA stability in *Drosophila* cells. Moreover, as in Neurospora and mammals (Zhou et al., 2016 PNAS; Fu et al., 2018 JBC). codon usage was also recently demonstrated to affect transcription in *Drosophila* cells by influencing chromatin modifications (Yang et al., 2021 NAR). These mechanisms can help explain the observed results.

3. Determination of tissue-specific expressed genes. The methods stated that "We defined genes as being tissue-specific in *Drosophila* if they were present in the 40% highest expressed genes in the tissue of interest and excluded from the 25% highest expressed genes of all other tissues analyzed." It's unclear how exactly these genes were selected. Does this mean that genes with low mRNA levels but highly tissue-specific were excluded from the analysis? If so, how was this justified? Tissue-specific genes are normally thought to be genes whose expression levels are much higher in minority of tissues than those in other tissues. Are mRNA fold-changes a better approach to determine tissue-specific genes? In the main text, it is stated that "we then took the 300 most enriched genes for each of 12 adult tissues". What does "most enriched" mean? mRNA levels or fold-change? Rationale to select 300 but not 500?

4. Figure 4. It is unclear why a new metrics needs to used. I suggest the authors also perform the analysis using the commonly used CAI or tAI.

5. Tissue-specific proteome data determined by quantitative mass spectrometry should be available, at least for the human/mice tissues. Does the analysis using protein data also lead to the same conclusion?

6. Figure 1E. Although the codon optimization increased protein level in the ovary, the effect is somewhat modest, considering the endogenous gene is highly specific for testis. Are other tissues also examined? Does the Endo reporter shown highly specific testis-specific expression? Was mRNA level also determined? It was recently shown in *Drosophila* cells that the codon usage effect on mRNA expression level is promoter-dependent (Yang et al., 2021 NAR). Under some promoters, codon usage has only weak effects on gene expression due to transcriptional effect. It is possible that the ubi promoter is one such promoter. This may also explain the effect seen in Figure 1.

*Reviewer #3 (Recommendations for the authors):*

1) Total mRNA (and perhaps nuclear mRNA) levels should be compared for the different reporters in whole larvae. This will start to provide insights into potential mechanisms that regulate the tissue specific expression pattern of rare codon containing genes. Without these data the authors should be cautious about any claims regarding the mode of differential regulation and whether this occurs at the level of transcription, mRNA stability, mRNA translation etc.

2) What cells in the ovary express RpL10Aa-com ? Do the ovaries of RpL10Aa-com females exhibit any overt phenotypes? Does expression of RpL10Aa in the ovary result in changes in RpL10Ab expression?

3) Please include details regarding the ubiquitin promoter and the 3'UTR used to modify the pBID-UASC vector. When assayed, qRT-PCR is used to quantify RNA levels. Are the authors confident that the same length transcript is being produced across different tissues for each of the reporters? Any differences could complicate the interpretation of the results.

4) Do the authors have an explanation for why GFP0D is not expressed in the female germline? Based on the image presented, most of the expression appears in the follicle cells.

5) In Figure 1—figure supplement 1C, the GFP band in 50C3' appears shifted up relative to the other lanes. Any differences in the protein product produced by a given reporter could complicate the interpretations of the results. In addition, there are big differences in protein loading in some of the lanes. The authors should provide a blot with more equal loading.

6) In Figure 3—figure supplement 1, many extra bands appear in the various lanes, especially in the testis and brain lanes of mGFP100Dv1. Do the authors have an explanation for these bands?

7) The authors should consider generating another RpL10Aa-com transgene with a point mutation that disrupts protein production. This would control for unanticipated effects caused by the presence of the RNA in the ovary and would strength the conclusion that production of RpL10Aa protein is detrimental to female fertility.

---

## [Author Response]

Essential revisions:All three reviewers agreed that the reported results are significant in providing evidence that differences in codon optimality in mRNAs can underlie tissue-specific differences in expression and that this phenomenon operates in restricting expression of an evolutionarily young endogenous gene to the testis versus ovaries in a manner important for female fertility in the fruit fly. At the same time, each noted that the underlying molecular mechanisms have not been probed, and that the scientific quality of the work would be enhanced considerably by additional experiments to determine whether the differences in expression arise primarily at the translational level or also/rather involve altered mRNA turnover or transcription in response to poor codon usage outside of testis and brain. To this end, the authors are requested to (i) provide measurements of reporter mRNA abundance in parallel with their measurements of reporter protein expression throughout the work;

As requested, we now include both protein and mRNA abundance throughout the manuscript (previously this was only provided in Figure 3). Specifically, revised Figure 1 now includes new data on mRNA abundance for all GFP reporters in panel S, matched alongside our previous protein data for these same reporters in panel R. Revised Figure 5 now includes new data on mRNA abundance for all RpL10Aa reporters in panel J, now matched with both previous and new data on protein abundance for these same reporters in panel E.

(ii) establish that the lengths of the mGFP100DV1 and RPL10a_endo reporter transcripts do not vary between testis and ovary;

Thank you for this question regarding mRNA lengths in different tissues. Alternate processing of UTRs, particularly the 5’UTR, could differ between tissues, and we agree that this is an important variable that should be included in our final manuscript. As requested, we now provide information on the 5’ and 3’UTR of our constructs, as well as new data regarding an analysis of 5’UTR processing in the testis and ovary.

First, we now add an entirely new supplemental figure (Figure 3 Figure Supplement 4), which contains the following:

A. An annotated map of the pBID vector used for all of our transgenic constructs, noting that the 5’UTR region is from the Ubip63E gene, and noting that the 3’UTR is a partial sequence of a Gypsy transposon insulator.

B. A diagram of the four known distinct 5’UTRs of the Ubip63E gene

C. New RT_PCR data from testis and ovary for mGFP100DV1, showing that we can detect two of the four alternatively spliced forms of the 5’UTR in both testis and ovary (we note that in both tissues one isoform was more readily detectable than the other, and the difference between these two isoforms is only 86 base pairs).

We also add new Table S3, reporting the 5’ UTR sequences recovered from testis and ovary.

In the revised manuscript, we now describe Figure Supplement 4 in the text related to Figure 3. We also note that we confirmed the sequence of the 3’UTR using 3’RACE, also reported in new table S2. We also detail our methods used to analyze the 5’ and 3’UTRs in the methods section of our revised manuscripts. We note that we did not directly examine mRNA length of RpL10Aa Endo, as this construct was cloned into the identical vector as all other constructs, and therefore we do not anticipate differential UTR processing between tissues for this inserted sequence. However, as detailed in our response to major reviewer question (iii), we do include new mRNA expression data for RPL10Aa Endo in the revised manuscript. Taken together, these reviewer-requested experiments improve the manuscript by adding information about mRNA length, and suggest that mRNA length differences from our transgenes are unlikely to be driving tissue-specific differences that we report.

(iii) verify that the Rpl10A_endo reporter is expressed selectively in testis by examining its expression in other tissues beside testis and ovaries. In addition to performing these additional experiments/analyses, the authors should address all of the other comments of the referees by adding the appropriate additional analyses or making the necessary revisions of text or figures. In those instances where a follow-up experiment has been proposed (in addition to those listed above), but you elect not to conduct it, it may be necessary for you to modify the text to acknowledge that its absence is a shortcoming of the current work."

As requested, we now provide additional clarifying information and several pieces of new data regarding FLAG-RpL10Aa_endo expression. As this is construct has an ectopic reporter/chromatin context, we do not necessarily expect its expression to be restricted to the testis. Along these lines, in the previous version of the manuscript, we showed that ectopically driven FLAG-RpL10Aa_endo protein is found in the ovary. Further, as requested by reviewers, we add new data on FLAG-RpL10Aa_endo protein expression in two settings beyond the testis and ovary, namely the gut and head (see revised Figure 5—figure supplement 1). This reviewer comment also pointed out to us that we should emphasize that our data on RpL10Aa rare codons really speak to function and not necessarily to restricted expression (given our use of an ectopic expression system). We have revised/clarified the text accordingly.

Further, and in line with reviewer major point (i), we did not provide information on testis protein expression for FLAG-RpL10Aa_endo in the previous manuscript. In the revised manuscript, we now add new data showing the relative protein expression of FLAG-RpL10Aa_endo and FLAG-RpL10Aa_common in the testis, paired with our previous data on protein expression of these constructs in the ovary (see revised Figure 5E). Additionally, as requested under major reviewer comment (i), we pair these data with new data showing the relative mRNA expression for RpL10Aa_endo and RpL10Aa_common in both the ovary and testis (see revised Figure 5J). Overall, our new data strengthen the conclusion that codon optimization of RpL10Aa increases its expression, and in the ovary this optimization causes female sterility.

Reviewer #1 (Recommendations for the authors):– Perhaps the authors would consider inverting the scale for the new taCAI metric so that the values would be lower than average, rather than higher than average, for tissues where the most abundant mRNAs tend to have low versus high CAI values.

We agree. We made this requested change (see revised Figure 4 and Figure 4—figure supplement 1).

– It's unclear from the experiments presented whether it is the number of rare codons or the proportion of rare codons in an mRNA that dictates the repression of expression above the 50% threshold in their reporters. It seems easier to imagine that it is the number of rare codons encountered during translation of an mRNA that contribute cumulatively to translational repression/mRNA decay. If so, short mRNAs with a high percentage of rare codons might be expressed above the 50% threshold because they can be completely translated without encountering the threshold number of rare codons. Can they address this point bioinformatically by examining native mRNAs, eg. to recalculate Figure 1S for short vs long mRNAs? Are there any studies in the literature that would allow the authors to comment on this issue in the Discussion?

As requested, we recalculated Fig1T for short vs. long mRNAs. This new analysis (see revised Figure 1—figure supplement 2) revealed no correlation between mRNA length and CAI for *Drosophila*. We similarly added a new analysis (see revised Figure 1—figure supplement 2) where we examined CAI by encoded protein length using a different binning strategy (as defined by Duret et al., 1999). Again, this separate analysis revealed no correlation between mRNA length and CAI for *Drosophila*. Additionally, as requested, we add to the revised manuscript a discussion of the study by Duret et al., (1999), suggesting that in some species there may be a correlation between long mRNAs and abundant rare codons. Separate from the above points, our data in Figure 2 show that both the GFP only reporter GFP54C3’ and the hybrid mCherry/GFP reporter mGFP100Dv1 achieve tissue specificity. This similar result with different reporters is despite GFP54C3’ being roughly half the size and a having a very different total abundance of rare codons compared to mGFP100Dv1. We share the reviewer’s interest in a future drilling down to essential sequence parameters underlying tissue specificity of rare codon expression. We look forward to pursuing these questions in future studies.

– If the protein expression data in Figure 3G were normalized for the mRNA levels in Figure 3N to obtain a measure of translation efficiency (TE), they could conclude that the TEs of the reporter are dramatically reduced in the ovary and AG, but comparable in the testis and brain. This in turn would suggest that impaired translation predominates, and increased mRNA turnover merely contributes, to the failure to express reporters exceeding 50% rare codons. Perhaps this would be a useful interpretation to consider adding to the paper.

Thank you for this helpful suggestion. As requested, we added new analysis to revised Figures 3O, which calculates the average translation efficiency in four different tissues. We note that this was aided by our addition of new data provided in response to major reviewer comment (i).

– Could the authors try to improve their presentation of the analysis in Figure 5A to make it easier to follow what they have done, and what they conclude from the difference in the distributions of data points in the 2^nd^ and 3^rd^ columns, and whether this difference is statistically significant.

In response, we replaced old Figure 5A with a new analysis heat map that should be easier to interpret. This change enabled us to simplify the corresponding results text. Thank you for this important feedback on figure clarity.

– It would be worth comparing the reporter mRNA levels to complement the protein expression data in Figure 5E to determine whether the increased expression of the Com reporter in ovaries occurs primarily at the level of translation with minimal contribution from increased mRNA stability.

Thank you for this comment- please see our response major point #1, where we addressed this point with new data.

Reviewer #2 (Recommendations for the authors):1. Figure 2. Did the authors try to check the tissue-specific expression of the series of reporters used Figure 1? One would predict that some of reporters should also show testis/brain-specific expression.

Thank you for the opportunity to clarify our methods with regards to Figure 1. We used a visual screen to check expression of each reporter, followed by whole animal western blots. For 11 of these 16 reporters, even the whole animal lacked detectable protein expression. Of the remaining 5, 3 had robust whole animal expression. This left two reporters where we did not detect robust tissue fluorescence in our visual screen. Rather than focus more on these constructs, we used this information to design the constructs in Figure 2, which became the focus of the rest of our study. To clarify our approach, we have revised the methods section.

2. Figure 2N. The Y axis abelling is hard to understand. Suggest to change to “Relative GFP mRNA level”. Are the mRNA differences between the two reporters in different tissues due to different mRNA decay rates? In addition to Burrow 2018 study, Zhao et al., (2017 NAR) also demonstrate an impact of codon usage on mRNA stability in *Drosophila* cells. Moreover, as in Neurospora and mammals (Zhou et al., 2016 PNAS; Fu et al., 2018 JBC). codon usage was also recently demonstrated to affect transcription in *Drosophila* cells by influencing chromatin modifications (Yang et al., 2021 NAR). These mechanisms can help explain the observed results.

As requested, we made the indicated label axis change (we believe the reviewer was referring to a panel in Figure 3, now found as revised Figure 3N). Additionally, we added the important and relevant citations mentioned by the reviewer. Further, we add new data to revised Figure 3—figure supplement 3, which in part speaks to the potential impact of chromatin (and therefore indirectly transcription) on our observed tissue specificity. Specifically, these new data reveal that we observe the same testis/brain specificity when we integrate a construct with identical codon content as mGFP100Dv1 at a different genomic location.

3. Determination of tissue-specific expressed genes. The methods stated that "We defined genes as being tissue-specific in *Drosophila* if they were present in the 40% highest expressed genes in the tissue of interest and excluded from the 25% highest expressed genes of all other tissues analyzed." It's unclear how exactly these genes were selected. Does this mean that genes with low mRNA levels but highly tissue-specific were excluded from the analysis? If so, how was this justified? Tissue-specific genes are normally thought to be genes whose expression levels are much higher in minority of tissues than those in other tissues. Are mRNA fold-changes a better approach to determine tissue-specific genes? In the main text, it is stated that "we then took the 300 most enriched genes for each of 12 adult tissues". What does "most enriched" mean? mRNA levels or fold-change? Rationale to select 300 but not 500?

Thank you for the opportunity to clarify our selection of mRNAs for our TA-CAI analysis. Regarding this question on mRNA abundance, we first wish to point out that our criteria to define an mRNA as tissue specific resulted in many tissues having less than 400 such genes. Hence, 500 genes was not possible and we settled on 300. We chose a thresholding, rather than a fold change approach, to determine tissue specificity. This was because some tissues have a much higher raw expression value than others (the median value is more than double for one tissue vs. another tissue in some cases), which confounds fold change approaches. We revised the text to expand on these important points for mRNA selection criteria.

4. Figure 4. It is unclear why a new metrics needs to used. I suggest the authors also perform the analysis using the commonly used CAI or tAI.

As requested, we computed tAI for the same dataset in Figure 4 (now included as a new analysis in revised Figure 4—figure supplement 1). This analysis did not reveal any tissue-specific differences. We actually expected this to be the case, as tAI reflects tRNA gene copy number, which is identical for all tissues in the same organism. We can understand that there are many metrics out there and there should be a good justification to add a new one. To make this point clearer, we added additional justification of our development of taCAI to the revised text.

5. Tissue-specific proteome data determined by quantitative mass spectrometry should be available, at least for the human/mice tissues. Does the analysis using protein data also lead to the same conclusion?

As requested, we add new analysis in new Figure 4—figure supplement 3 of tissue-specific quantitative mass spectrometry data from Jiang et al., 2020 from 19 human tissues for which there were at least 10 abundant proteins (we note that our need to use such a low cut-off reflects a caveat to the ability to detect the abundance of many tissue-specific proteins by quantitative mass spectrometry). Despite caveats with these data, our analysis supports our RNAseq-based conclusions, namely that the testis has an abundance of proteins with low CAI. Thank you to the reviewer for this suggestion.

6. Figure 1E. Although the codon optimization increased protein level in the ovary, the effect is somewhat modest, considering the endogenous gene is highly specific for testis. Are other tissues also examined? Does the Endo reporter shown highly specific testis-specific expression? Was mRNA level also determined? It was recently shown in *Drosophila* cells that the codon usage effect on mRNA expression level is promoter-dependent (Yang et al., 2021 NAR). Under some promoters, codon usage has only weak effects on gene expression due to transcriptional effect. It is possible that the ubi promoter is one such promoter. This may also explain the effect seen in Figure 1.

As requested, and as discussed above for major reviewer point (iii), we now add new data showing FLAG-RpL10Aa_endo protein expression in four total tissues. These new data show that our exogenous reporter expression is not specific to the testis. From these data, we conclude that even a modest codon-dependent increase in RpL10Aa protein in the ovary has detrimental impacts on female fertility. As discussed below for reviewer 3- comment #2, we add new data to revised Figure 5, which further reveals that the modest codon-dependent increase in RpL10Aa protein expression causes massive disruption of oogenesis.

Further, as requested and as already discussed above for major reviewer point (i), we now add new data to Figure 5 on mRNA measurements in the testis and ovary of RpL10Aa endo and RpL10Aa common. We note in the revised text that the way we performed these measurements for RpL10Aa endo will simultaneously detect both endogenous and transgenic RpL10Aa. Indeed, we detect a much higher level of RpL10Aa endo in the testis relative to RpL10Aa common, likely reflecting the abundance of non-transgenic RpL10Aa endo in the testis. This trend (more endo than common) is not seen in the ovary. When we take all of our data on mRNA and protein expression together for RpL10Aa, it supports a model whereby common codons from an exogenous transgene cause a modest increase in ovarian expression of RpL10Aa, taking it above a threshold of expression that correlates with massive disruption of oogenesis and female fertility. We expand our discussion of these results in the revised text.

Despite the massive phenotypic differences that we observe between RpL10Aa_endo and RpL10Aa_common, the reviewer is correct that the expression differences are modest. And you are likely correct that the promoter may underlie some of the modest differences.

Reviewer #3 (Recommendations for the authors):1) Total mRNA (and perhaps nuclear mRNA) levels should be compared for the different reporters in whole larvae. This will start to provide insights into potential mechanisms that regulate the tissue specific expression pattern of rare codon containing genes. Without these data the authors should be cautious about any claims regarding the mode of differential regulation and whether this occurs at the level of transcription, mRNA stability, mRNA translation etc.

As requested, we provided these new data as part of our response to major reviewer request (i). We also reviewed the manuscript and made several revisions (see our tracked changes) to make sure we are cautious about any claims regarding the numerous possible RNA regulatory mechanisms that could or could not be in play.

2) What cells in the ovary express RpL10Aa-com ? Do the ovaries of RpL10Aa-com females exhibit any overt phenotypes? Does expression of RpL10Aa in the ovary result in changes in RpL10Ab expression?

Thank you for the opportunity to clarify regarding expression of RpL10Aa-com. This exogenous construct was expressed from the same Ubi promoter as our GFP constructs. Per your subsequent comment (reviewer 3 comment #4), we provide new data showing that at all stages of oogenesis that we examined, both follicle cells and nurse cells clearly express Ubi-driven transgenes, though the somatic follicle cells express such transgenes at a higher level (Figure 3 Figure Supplement 1C). Therefore, we conclude that RpL10Aa com is expressed in both somatic and germline cells. We have added new discussion of this expression pattern to the revised text.

Regarding phenotypes of the sterile animals, in response we now provide new data in revised Figure 5G-I, showing that stage 8-10 ovaries expressing RpL10Aa-com exhibit a massive increase in pyknotic nuclei, whereas earlier stage ovaries appear normal. This is consistent with activation of a mid-oogenesis apoptosis checkpoint. We have added discussion of these data to the revised manuscript.

With regards to RpL10Ab expression, we attempted to obtain a previously generated antibody to this protein from Dr. Silke Dorner. However, Dr. Dorner has apparently closed their lab. We agree that this is an interesting question for future analysis.

3) Please include details regarding the ubiquitin promoter and the 3'UTR used to modify the pBID-UASC vector. When assayed, qRT-PCR is used to quantify RNA levels. Are the authors confident that the same length transcript is being produced across different tissues for each of the reporters? Any differences could complicate the interpretation of the results.

As requested, we performed the necessary assays and provided the new data and requested information as detailed under major reviewer request (ii). Please also see revised Table S3 for the details regarding the 5’ and 3’ UTRs.

4) Do the authors have an explanation for why GFP0D is not expressed in the female germline? Based on the image presented, most of the expression appears in the follicle cells.

As requested, we provide new data regarding GFP0D expression in the ovary in revised Figure 3—figure supplement 1C, showing that this construct is in fact expressed in the female germline, though at reduced levels relative to the follicle cells of the soma. We have added discussion of this difference to the revised text.

5) In Figure 1—figure supplement 1C, the GFP band in 50C3' appears shifted up relative to the other lanes. Any differences in the protein product produced by a given reporter could complicate the interpretations of the results. In addition, there are big differences in protein loading in some of the lanes. The authors should provide a blot with more equal loading.

Thank you for this comment. We agree with the reviewer that any differences for these two reporters (50C3’ and 50C5’) could complicate the interpretations of the results in this specific instance. As these constructs did not play a part in any of our main conclusions, we elected to not repeat this experiment. We instead note in the revised text the caveats that pertain only to these two constructs.

6) In Figure 3—figure supplement 1, many extra bands appear in the various lanes, especially in the testis and brain lanes of mGFP100Dv1. Do the authors have an explanation for these bands?

Thank you for the opportunity to discuss extra bands of unexpected size. In response, we have revised the discussion to discuss possible tissue-specific processing of protein in this supplemental figure data.

7) The authors should consider generating another RpL10Aa-com transgene with a point mutation that disrupts protein production. This would control for unanticipated effects caused by the presence of the RNA in the ovary and would strength the conclusion that production of RpL10Aa protein is detrimental to female fertility.

We agree with the reviewer that this would be an interesting control. Per the editor’s instructions, we elected to not perform this experiment. Instead, we revised the text to discuss possible unanticipated effects caused by RpL10Aa common RNA expression in the ovary.